# CCR4 and CCR7 differentially regulate thymocyte localization with distinct outcomes for central tolerance

Yu Li[1†], Pablo Guaman Tipan[1†], Hilary J Selden[1], Jayashree Srinivasan[1], Laura P Hale[2], Lauren IR Ehrlich[1,3]*

[1]Department of Molecular Biosciences, The University of Texas at Austin, Austin, United States; [2]Department of Pathology, Duke University, Durham, United States; [3]Department of Oncology, LIVESTRONG Cancer Institutes, The University of Texas at Austin Dell Medical School, Austin, United States

*For correspondence: lehrlich@austin.utexas.edu

†These authors contributed equally to this work

Competing interest: The authors declare that no competing interests exist.

**Abstract** Central tolerance ensures autoreactive T cells are eliminated or diverted to the regulatory T cell lineage, thus preventing autoimmunity. To undergo central tolerance, thymocytes must enter the medulla to test their T-cell receptors (TCRs) for autoreactivity against the diverse self-antigens displayed by antigen-presenting cells (APCs). While CCR7 is known to promote thymocyte medullary entry and negative selection, our previous studies implicate CCR4 in these processes, raising the question of whether CCR4 and CCR7 play distinct or redundant roles in central tolerance. Here, synchronized positive selection assays, two-photon time-lapse microscopy, and quantification of TCR-signaled apoptotic thymocytes, demonstrate that CCR4 and CCR7 promote medullary accumulation and central tolerance of distinct post-positive selection thymocyte subsets in mice. CCR4 is upregulated within hours of positive selection signaling and promotes medullary entry and clonal deletion of immature post-positive selection thymocytes. In contrast, CCR7 is expressed several days later and is required for medullary localization and negative selection of mature thymocytes. In addition, CCR4 and CCR7 differentially enforce self-tolerance, with CCR4 enforcing tolerance to self-antigens presented by activated APCs, which express CCR4 ligands. Our findings show that CCR7 expression is not synonymous with medullary localization and support a revised model of central tolerance in which CCR4 and CCR7 promote early and late stages of negative selection, respectively, via interactions with distinct APC subsets.

## Editor's evaluation

This important paper reveals the key steps associated with intrathymic central tolerance. Using elegant live imaging approaches, the authors provide convincing evidence in support of an updated model for how positive-selected thymocytes are called into the thymus medulla to interact with distinct antigen-presenting cells. The work makes an important contribution to the field by identifying previously unappreciated complexities related to cellular movement during T cell generation.

## Introduction

Self-tolerance of the T cell repertoire is established in thymus through the process of central tolerance, which encompasses both negative selection and regulatory T cell (Treg) induction (*Klein et al., 2014*). To avoid autoimmunity, developing T cells must be broadly tolerized not only to ubiquitously expressed self-antigens, but also to proteins expressed by distinct differentiated cell types. The thymic medulla is a specialized environment in which central tolerance to such diverse self-antigens

**eLife digest** Autoimmune diseases occur when immune cells mistakenly identify the body's own tissues as 'foreign' and attack them. To reduce the risk of this happening, the body has multiple ways of removing self-reactive immune cells, including T cells. One such way, known as central tolerance, occurs in the thymus – the organ where T cells develop.

In the center of the thymus – the medulla – specialized cells display fragments of the majority of proteins expressed by healthy cells throughout the body. Developing T cells enter the medulla, where they scan these specialized cells to determine if they recognize the presented protein fragments. If an immature T cell recognizes and binds to these 'self-antigens' too strongly, it is either destroyed, or it develops into a regulatory cell, capable of actively suppressing T cell responses to that self-antigen. This ensures that T cells won't attack healthy cells in the body that make those self-antigens, and therefore, it is important that T cells enter the medulla and carry out this scanning process efficiently.

T cells are recruited to the medulla from the outer region of the thymus by chemical signals called chemokines. These signals are recognized by chemokine receptors on T cells, which are expressed at different times during T cell development. Previous work has shown that one of these receptors, called CCR7, guides T cells to the medulla. Although it was thought that CCR7 was solely responsible for this migration, prior work suggests another receptor, CCR4, may also contribute to T cell migration into the medulla and central tolerance.

To determine whether CCR7 and CCR4 play the same or different roles in central tolerance, Li, Tipan et al. used a combination of experimental methods, including live imaging of the thymus, to study T cell development in mice. The experiments revealed that CCR4 is expressed first, and this receptor alone guides immature T cells into the medulla and ensures that they are the first to be checked for self-reactivity. In contrast, CCR7 is expressed by more mature developing T cells two to three days later, ensuring they also accumulate within the medulla and become tolerant to self-antigens. Both receptors are required for protection from autoimmunity, with results suggesting that CCR4 and CCR7 promote tolerance against different tissues.

Taken together, the findings provide new information about the distinct requirement for CCR4 and CCR7 in guiding immature T cells into the medulla and ensuring central tolerance to diverse tissues. One outstanding question is whether defects in T cells entering the medulla earlier or later alter tolerance to distinct self-antigens and lead to different autoimmune diseases. Future work will also investigate whether these observations hold true in humans, potentially leading to therapies for autoimmune diseases.

is enforced. After positive selection in the cortex, thymocytes migrate into the medulla, where they interact with medullary antigen-presenting cells (APCs), including thymic dendritic cells (DCs) and medullary thymic epithelial cells (mTECs). Collectively, these APCs enforce self-tolerance by presenting self-peptides from the majority of the proteome on MHC complexes, such that thymocytes expressing autoreactive T cell receptors undergo clonal deletion or diversion to the Treg lineage (*Ehrlich, 2016*; *Klein et al., 2014*; *Klein et al., 2019*; *Lancaster et al., 2019*). AIRE[+] mTECs express >80% of the proteome, including *Aire*-dependent tissue restricted antigens (TRAs) that are otherwise expressed in only a few peripheral tissues (*Abramson and Anderson, 2017*; *Anderson and Su, 2016*; *Mathis and Benoist, 2007*; *Meredith et al., 2015*). DCs also present numerous self-antigens, including those acquired from mTECs, from circulation, or from peripheral tissues and trafficked into the thymus (*Atibalentja et al., 2009*; *Bonasio et al., 2006*; *Koble and Kyewski, 2009*). The importance of inducing thymocyte tolerance to medullary self-antigens is evidenced by multi-organ autoimmunity that ensues in *Aire*-deficient mice and APECED patients (*Anderson et al., 2002*; *Finnish-German APECED Consortium, 1997*; *Nagamine et al., 1997*), in whom thymic expression of TRAs is greatly diminished. Failure to express even a single medullary TRA can result in impaired tolerance and subsequent T cell-mediated autoimmune pathology (*DeVoss et al., 2006*). Notably, individual TRAs are expressed by only 1–3% of *Aire*[+] mTECs (*Brennecke et al., 2015*; *Meredith et al., 2015*; *Sansom et al., 2014*), resulting in a sparse mosaic of self-antigen display throughout the medulla. Because thymocytes reside in the medulla for only 4–5 days (*McCaughtry et al., 2007*), tight spatiotemporal regulation is required to ensure that post-positive

selection thymocytes encounter the full spectrum of medullary self-antigens required to enforce self-tolerance.

To facilitate medullary entry after positive selection, thymocytes upregulate chemokine receptors that promote their directional migration toward medullary biased chemokine gradients (*Bleul and Boehm, 2000*; *Hu et al., 2015a*; *Lancaster et al., 2018*; *Petrie and Zúñiga-Pflücker, 2007*). Notably, CCR7 has been shown to play a critical role in directing chemotaxis of post-positive selection thymocytes toward the medulla, where CCR7 ligands are expressed, enhancing thymocyte accumulation within the medulla, enforcing negative selection to TRAs, and averting autoimmunity (*Ehrlich et al., 2009*; *Kozai et al., 2017*; *Kurobe et al., 2006*; *Nitta et al., 2009*; *Ueno et al., 2004*). Given the significance of CCR7 in promoting thymocyte medullary entry, CCR7 expression is widely considered to be synonymous with thymocyte medullary localization. Based largely on this definition, recent studies indicate that despite the well-established role of the medulla in inducing central tolerance, ~75% of negative selection occurs in CCR7$^-$ 'cortical' thymocytes, while only 25% occurs in CCR7$^+$ 'medullary' cells (*Breed et al., 2019*; *Daley et al., 2013*; *Hu et al., 2016*). It has also been suggested that cortical negative selection may eliminate thymocytes reactive to ubiquitous self-antigens while medullary deletion tolerizes TRA-responsive cells. Our previous research implicates chemokine receptors other than CCR7 in promoting thymocyte medullary localization (*Ehrlich et al., 2009*). We found that CCR4, which is upregulated by post-positive selection thymocytes, also promotes medullary entry and negative selection (*Hu et al., 2015b*). These findings raise the question of whether CCR4 and CCR7 play distinct or redundant roles in promoting thymocyte medullary entry and central tolerance.

In this paper, we use a combination of approaches, including chemotaxis assays, two-photon live-cell microscopy, and synchronized positive selection assays, to distinguish the contributions of CCR4 versus CCR7 to thymocyte medullary entry and negative selection. We find that CCR4 is upregulated by thymocytes as early as a few hours after positive selection, while CCR7 is expressed days later by mature post-positive selection cells. CCR4 expression promotes chemotaxis and medullary entry of early post-positive selection thymocytes, which do not yet express CCR7. Initial CCR7 expression results in only moderate thymocyte chemotaxis toward CCR7 ligands and modest accumulation in the medulla, such that these thymocytes migrate in both the cortex and medulla. These findings indicate that CCR7 expression is not a definitive marker of medullary versus cortical localization. CCR7 is, however, required for robust accumulation of mature (CD4SP) thymocytes in the medulla. Notably, consistent with their differential activity in distinct thymocyte subsets, CCR4 and CCR7 are required in early and late phases of polyclonal negative selection, respectively. While CCR7 is known to promote tolerance to TRAs (*Kozai et al., 2017*; *Nitta et al., 2009*), we present evidence that CCR4 promotes tolerance to self-antigens presented by activated APCs. Collectively, our study establishes non-redundant roles for CCR4 and CCR7 in governing localization of post-positive selection thymocyte subsets and central tolerance.

## Results

### CCR4 is expressed by immature post-positive selection thymocyte subsets, while CCR7 expression is restricted to more mature thymocyte subsets

We first investigated expression of CCR4 and CCR7 by distinct thymocyte subsets using cell-surface markers that delineate developmental stages after positive selection (*Figure 1A*; *Sinclair et al., 2013*; *Xing et al., 2016*). To validate the developmental trajectory of these thymocyte subsets, we analyzed their GFP expression levels in Rag2p-GFP mice, in which GFP expression is driven by the *Rag2* promoter. In this model, expression of GFP distinguishes newly generated thymocytes (GFP$^+$) from recirculating T cells (GFP$^-$), and declining GFP expression reflects time elapsed after positive selection, when the *Rag2* promoter becomes inactive (*Boursalian et al., 2004*). In pre-positive selection CD4$^+$CD8$^+$ double-positive (DP) CD3$^-$CD69$^-$ thymocytes, the *Rag2* promoter is active, resulting in maximal GFP expression (*Figure 1B*). After receiving a TCR signal, thymocytes upregulate CD3 and CD69 (*Fu et al., 2009*), generating early post-positive selection DP CD3$^{lo}$CD69$^+$ cells; the brief time elapsed after positive selection is indicated by continued high GFP expression by this subset (*Figure 1B*). To further evaluate if DP CD3$^{lo}$CD69$^+$ thymocytes represent post-positive selection thymocytes, as opposed to cells undergoing strong TCR signaling driving clonal deletion, we evaluated

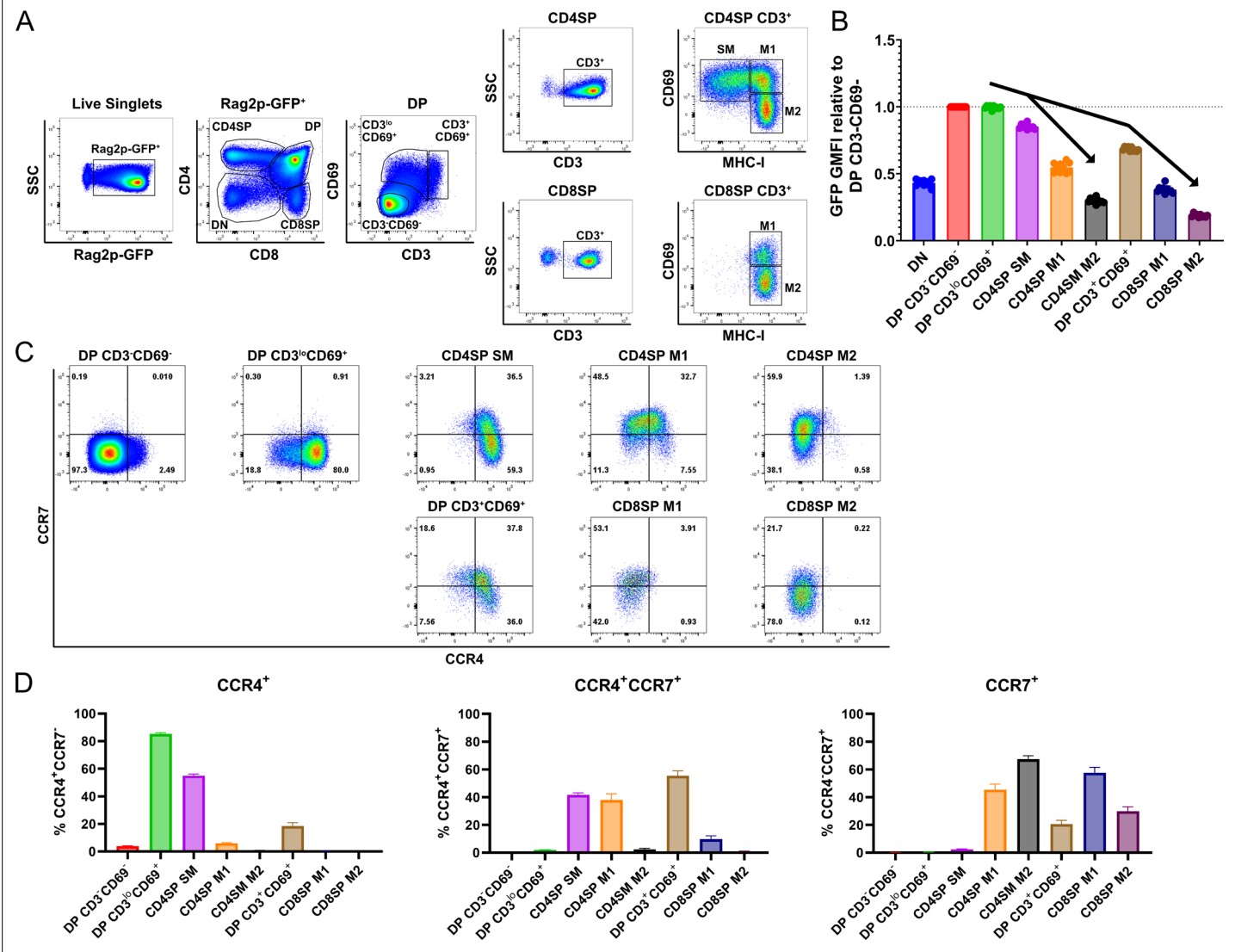

**Figure 1.** CCR4 and CCR7 are expressed by immature and mature subsets of post-positive selection thymocytes, respectively. (**A**) Flow cytometry gating scheme to delineate thymocyte subsets in Rag2p-GFP mice. Post-positive selection DP (CD3$^{lo}$CD69$^+$ and CD3$^+$CD69$^+$), CD4SP (SM, M1, and M2), and CD8SP (M1 and M2) subsets are annotated. (**B**) Relative Rag2p-GFP Geometric Mean Fluorescent Intensity (GMFI) of each subset normalized to pre-positive selection DP CD3$^-$CD69$^-$ cells. Individual data points represent average GFP GMFI of the indicated subset from each mouse. (**C**) Representative flow cytometry plots showing CCR4 and CCR7 expression by post-positive selection thymocyte subsets. (**D**) Quantification of the percentage of cells of the indicated subset that express CCR4, CCR7, or both chemokine receptors. For (**B**) and (**D**), data were compiled from four independent experiments (mean ± standard error of the mean [SEM]; N = 10 mice).

The online version of this article includes the following figure supplement(s) for figure 1:

**Figure supplement 1.** DP CD3$^{lo}$CD69$^+$ cells are consistent with post-positive selection DPs and DP CD3$^+$CD69$^+$ cells represent co-receptor reversing MHCI-restricted thymocytes.

**Figure supplement 2.** CCR7 expression on developing thymocytes begin at the CD4SP SM stage.

thymocyte subsets from *Nr4a1*$^{GFP}$ mice, in which GFP levels reflect the strength of TCR signaling (*Figure 1—figure supplement 1*; *Moran et al., 2011*). Elevation of GFP expression by DP CD3$^{lo}$CD69$^+$ cells, relative to DP CD3$^-$CD69$^-$ cells, confirmed this subset was TCR signaled. Thymocytes rescued from negative selection express *Nr4a1*$^{GFP}$ at levels comparable to thymic Tregs, which express self-reactive TCRs (*Stritesky et al., 2013*). GFP levels were substantially lower in DP CD3$^{lo}$CD69$^+$ cells than in CD25$^+$ CD4SPs, which are mostly Tregs (*Figure 1—figure supplement 1A*). Together, these data are consistent with DP CD3$^{lo}$CD69$^+$ cells representing mainly post-positive selection DPs, and not strongly self-reactive thymocytes undergoing negative selection. Sequential maturation through CD4$^+$

single-positive semi-mature cells (CD4SP SM) and then CD4SP or CD8SP mature 1 (M1) and mature 2 (M2) subsets was confirmed by progressive reductions in GFP levels for both the CD4 and CD8 lineages (*Figure 1B*; *Xing et al., 2016*). A distinct population of DP CD3+CD69+ cells expressed lower levels of GFP than CD4SP SM cells (*Figure 1B*), placing it temporally between CD4SP SM and CD8SP M1 cells (*Figure 1B*). Thus, we considered whether DP CD3+CD69+ cells represent MHC-I-restricted thymocytes in the process of differentiating into CD8SP cells through co-receptor reversal down-stream of the CD4SP SM stage (*Brugnera et al., 2000*). Consistent with this possibility, DP CD3+CD69+ cells were significantly reduced in $B2m^{-/-}$ mice, in which positive selection of CD8SP thymocytes is abrogated, as is selection of innate iNKT cells and CD8αα+ IELps (*Bendelac et al., 1994*; *Ruscher et al., 2017*). Furthermore, DP CD3+CD69+ cells were enriched in mice with a deletion spanning the MHC-II genes *H2-Aa*, *H2-Eb1*, and *H2-Eb2* ($MHCII^{-/-}$) in which only MHC-I-restricted thymocytes are positively selected (*Figure 1—figure supplement 1B, C*). Together, these data demonstrate that DP CD3+CD69+ cells are MHC-I restricted and have persisted for an intermediate time between CD4SP SM and CD8SP M1 cells after positive selection, consistent with a co-receptor reversing DP subset.

Having delineated the temporal sequence of thymocyte development post-positive selection, we quantified expression of CCR4 and CCR7 by each subset. CCR4 is upregulated by the majority of early post-positive selection DP CD3^lo^CD69+ thymocytes, which do not yet express CCR7 (*Figure 1C, D*, *Figure 1—figure supplement 2A, B*). CCR7 is subsequently expressed at high levels by ~40% CD4SP SM cells, and CCR4 expression persists on this subset (*Figure 1C, D*). In comparison to $Ccr7^{-/-}$ cells, low-level expression of CCR7 can be detected on additional $Ccr7^{+/+}$ CD4SP SM cells, consistent with initial upregulation of CCR7 by this subset (*Figure 1—figure supplement 2A, B*). Expression of CCR4 and CCR7 by MHC-I-restricted DP CD3+CD69+ cells is similar to that of CD4SP SM cells, with slightly more CCR7 and less CCR4 (*Figure 1C, D*, *Figure 1—figure supplement 2A, B*). As thymocytes mature through CD4SP and CD8SP M1 and M2 stages, CCR4 is progressively downreg-ulated. CCR7 expression peaks in both CD4SP and CD8SP M1 subsets and is diminished in M2 cells (*Figure 1C, D*, *Figure 1—figure supplement 2A, B*). Because cell-surface expression of CCR7 on mature CD4+ T cells can be downregulated due to ligand-induced internalization of the chemokine receptor (*Britschgi et al., 2008*), we tested whether receptor internalization could have obscured detection of CCR7 or CCR4 on ex vivo thymocyte subsets. Thymocytes were incubated at 37°C for up to 2 hr, to allow re-expression of internalized receptors, before immunostaining with anti-CCR7 and anti-CCR4 antibodies. Increased expression of cell-surface CCR7 or CCR4 was not detected with increasing incubation time (*Figure 1—figure supplement 2C, D*). These findings confirm that CCR4 expression is an early indicator of positive selection at the DP stage, while CCR7 expression is upreg-ulated later by more mature CD4SP and CD8SP subsets.

## Early post-positive selection thymocytes undergo chemotaxis toward CCR4 ligands, while more mature subsets respond to CCR7 ligands

We next determined whether expression of CCR4 and CCR7 corresponds to thymocyte chemotactic responses to their respective ligands. DP CD3^lo^CD69+ cells, which express CCR4 but not CCR7, under-went chemotaxis in response to the CCR4 ligand CCL22, but not the CCR7 ligands CCL19 or CCL21 (*Figure 2A–D*, *Figure 2—figure supplement 1A*). CCL22-induced chemotaxis of all subsequent post-positive selection thymocytes, except the most mature M2 thymocyte subsets (*Figure 2B*, *Figure 2—figure supplement 1A*), largely in keeping with CCR4 expression patterns. Although CD4SP SM and DP CD3+CD69+ subsets expressed CCR7 (*Figure 1C*), CCR7-mediated chemotaxis of these subsets did not reach statistical significance when compared to all thymocyte subsets (*Figure 2C, D*). However, when considered separately, moderate CCR7-mediated chemotaxis of these subsets was revealed, although DP CD3^lo^CD69+ cells still did not undergo CCR7-mediated chemotaxis (*Figure 2—figure supplement 1A*). Consistent with expression of CCR7 by more mature thymocyte subsets, the CCR7 ligands CCL19 and CCL21 induced highly efficient chemotaxis of mature CD4SP and CD8SP M1 and M2 subsets (*Figure 2C, D*).

A recent study indicated that expression of the chemokine receptor CXCR4 must be extinguished before post-positive selection thymocytes can leave the cortex, where CXCL12 is expressed by cortical thymic epithelial cells, to migrate into the medulla (*Kadakia et al., 2019*). Thus, we sought to place CXCR4 expression and function into context with CCR4 and CCR7. CXCR4 expression decreased after positive selection, but all post-positive selection subsets continued to express CXCR4 (*Figure 3A*,

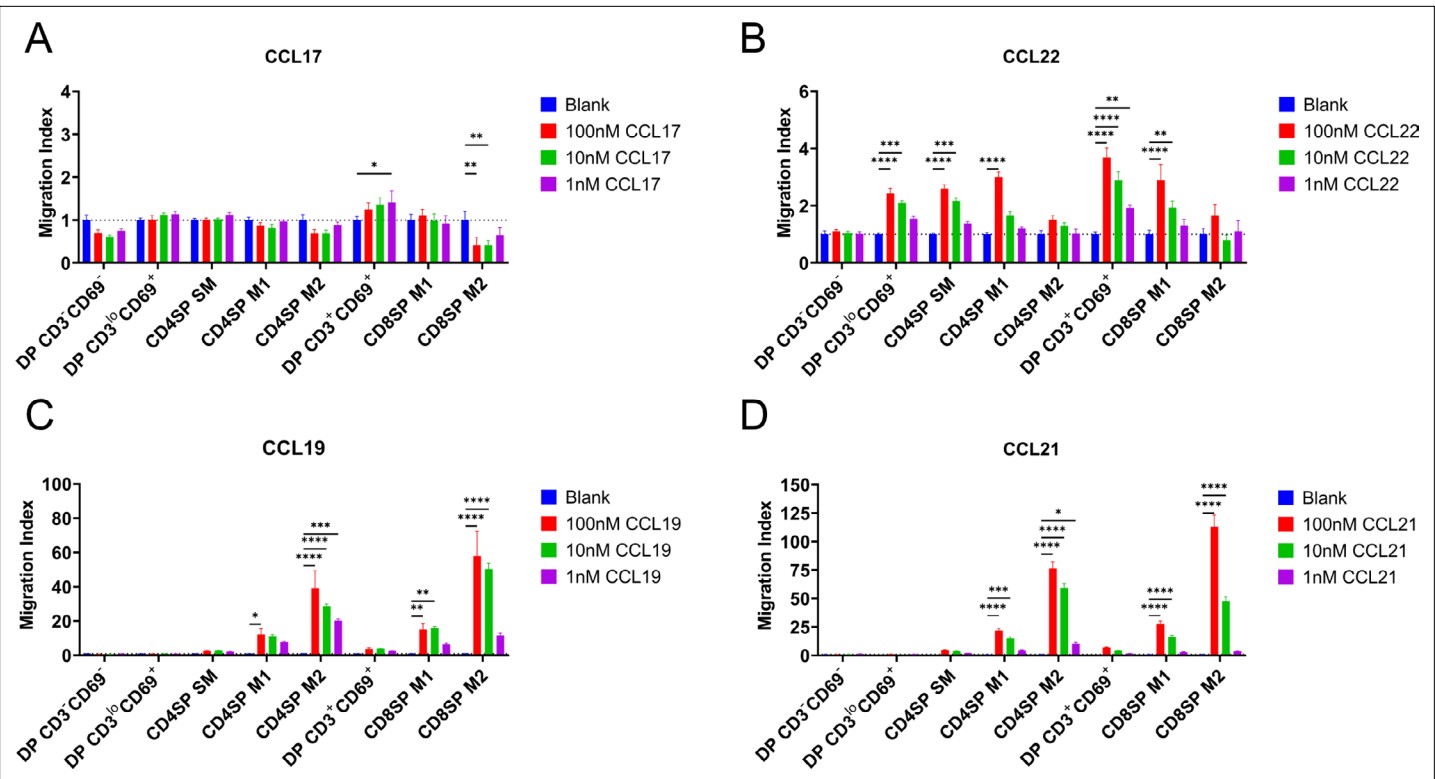

**Figure 2.** CCR4 and CCR7 promote chemotaxis of immature and mature post-positive selection thymocyte subsets, respectively. Transwell assays were used to quantify chemotaxis of thymocyte subsets to the indicated concentrations of the CCR4 ligands CCL17 (**A**) and CCL22 (**B**) and the CCR7 ligands CCL19 (**C**) and CCL21 (**D**). Migration index was calculated as the frequency of input cells of each subset that migrated toward the chemokine relative to the frequency of input cells that migrated in the absence of chemokine (Blank). Data were compiled from three independent experiments with triplicate wells per assay. Two-way ANOVA with Dunnett's multiple comparisons correction was used for statistical analysis (mean ± SEM; *p < 0.05, **p < 0.01, ***p < 0.001, ****p < 0.0001).

The online version of this article includes the following figure supplement(s) for figure 2:

**Figure supplement 1.** Chemokine signaling is not diminished in CD4SP SM thymocytes.

**Figure supplement 2.** CD4SP SM thymocytes migrate toward CCL25 and CCL22 is localized in the medulla.

*B*), and all except for CD4SP SM and DP CD3+CD69+ subsets underwent chemotaxis in response to CXCL12 (*Figure 3C*). Although the earliest post-positive selection thymocyte subsets exhibited reduced CXCR4 responsiveness, perhaps allowing them to exit the cortex, the ability to respond to CXCR4 signals does not preclude medullary localization, as evidenced by CXCR4-mediated chemotaxis of medullary CD4SP and CD8SP M2 cells. Taken together, chemotaxis assays show that early post-positive selection DP CD3loCD69+ thymocytes are responsive to CCR4 but not CCR7 ligands, with low responsiveness to CXCR4 ligands. At the next stage, CD4SP SM cells also undergo chemotaxis toward CCR4 ligands, and begin to migrate at a low level toward CCR7 ligands, but not to CXCR4 ligands. As thymocytes mature to the CD4SP and CD8SP M1 and M2 stages, they progressively gain the ability to respond robustly to CCR7 ligands, regain CXCR4 responsiveness, and lose responsiveness to CCR4 ligands (*Figure 3D*).

Initial expression of CCR7 by CD4SP SM thymocytes does not result in strong chemotaxis to CCR7 ligands (*Figure 2D*), and expression of CXCR4 by this subset does not enable detectable chemotaxis toward CXCL12 (*Figure 3C*), highlighting a temporal disconnect between chemokine receptor expression and function on maturing post-positive selection thymocytes. We considered the possibility that CD4SP SM cells could be intrinsically unable to respond to chemokine receptor signaling. However, these cells responded efficiently to CCR4 ligands (*Figure 2B*), and they underwent chemotaxis to the CCR9 ligand CCL25 at levels comparable to DP CD3loCD69+ cells (*Figure 2—figure supplement 2A*), consistent with a previous study showing CCL25 induces migration of most thymocyte subsets

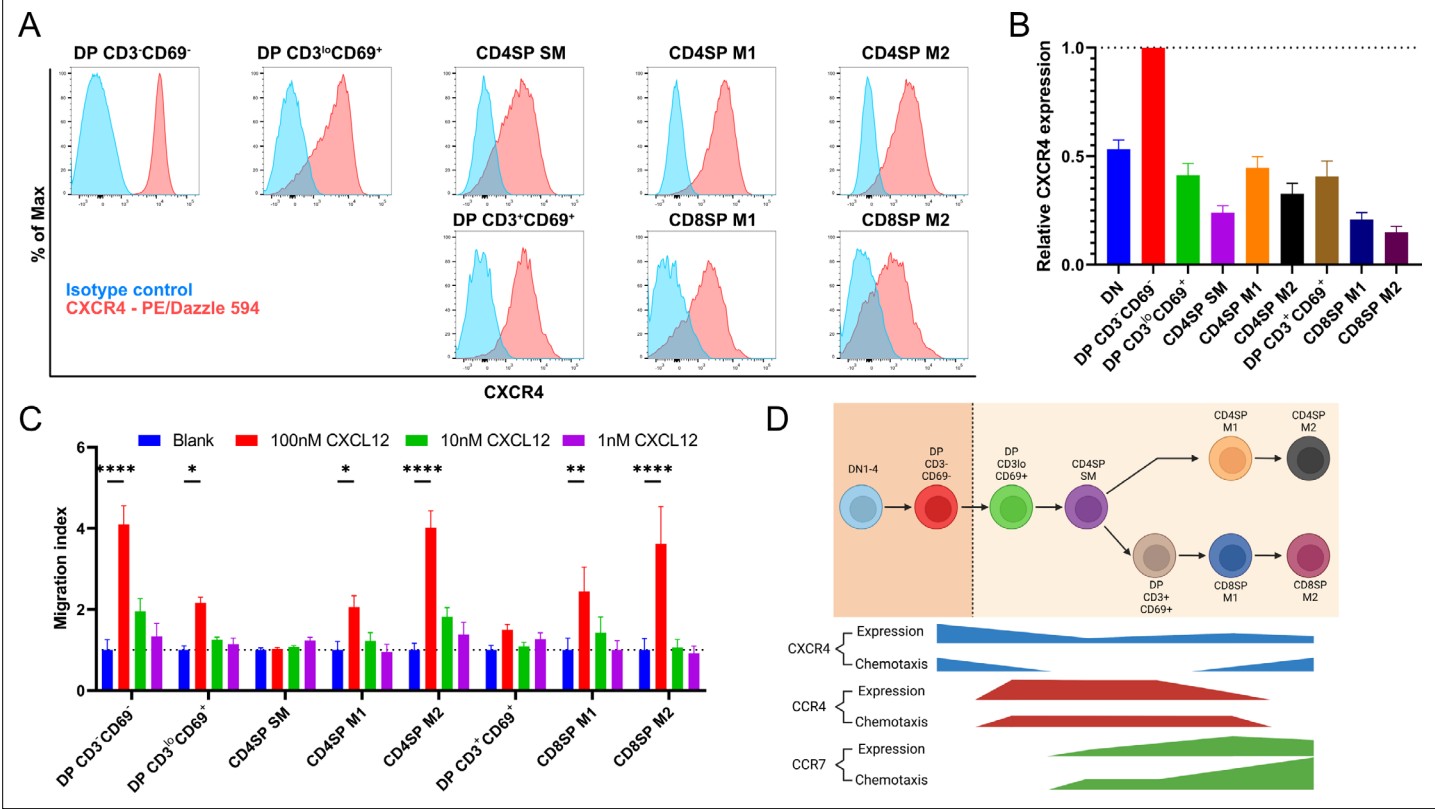

**Figure 3.** Persistent CXCR4 expression was detected on post-positive selection thymocytes, and CXCR4 activity declines only in intermediate subsets. (**A**) Representative flow cytometry plots showing CXCR4 surface expression by the indicated thymocyte subsets, compared to isotype control stains. (**B**) Quantification of relative GMFI of CXCR4 for the indicated thymocyte subsets (mean ± SEM; $N = 10$). (**C**) Transwell assays were used to quantify chemotaxis of thymocyte subsets to the indicated concentrations of CXCL12. Migration index was calculated as the frequency of input cells of each subset that migrated toward the chemokine relative to the frequency of input cells that migrated in the absence of chemokine (Blank). Data were compiled from three independent experiments with triplicate wells per assay. Two-way ANOVA with Dunnett's multiple comparisons correction was used for statistical analysis (mean ± SEM; *p < 0.05, **p < 0.01, ****p < 0.0001). (**D**) Graphical summary showing relative expression and chemotactic activity of CXCR4, CCR4, and CCR7 on post-positive selection thymocyte subsets based on data in *Figures 1–3*.

(*Campbell et al., 1999*). Thus, regulation of thymocyte responsiveness to chemokine receptors is more complex than altered cell-surface expression levels.

CCL21a establishes a chemokine gradient that drives accumulation of CD4SP thymocytes in the medulla (*Kozai et al., 2017*; *Ueno et al., 2002*; *Ueno et al., 2004*). Because we found that functional CCR4 is expressed by early post-positive selection DP thymocytes, we examined whether CCL22 could establish a similar chemokine gradient to recruit these CCR4-responsive cells into the medulla. Immunofluorescence analysis revealed that CCL22 is expressed predominantly in the medulla (*Figure 2—figure supplement 2B*), consistent with a potential role in inducing medullary entry of early post-positive selection thymocytes.

## The timing of CCR4 upregulation following positive selection correlates with thymocyte medullary entry

Given that the majority of early post-positive selection DP thymocytes respond to CCR4 but not CCR7 ligands, we sought to determine how rapidly CCR4 versus CCR7 are upregulated following initiation of positive selection and to assess whether expression of either receptor correlates with the timing of medullary entry. To accomplish these goals, we used a synchronized positive selection thymic slice assay, coupled with flow cytometry or two-photon microscopy (*Figure 4A*). OT-II⁺ *Rag2⁻/⁻ MHCII⁻/⁻* mice served as a source of pre-positive selection DP thymocytes that did not yet express CCR4 or CCR7 due to lack of positive selection in the absence of MHC-II expression (*Figure 4B*). OT-II⁺ *Rag2⁻/⁻ MHCII⁻/⁻* thymocytes were fluorescently labeled prior to culturing on MHC-II-sufficient,

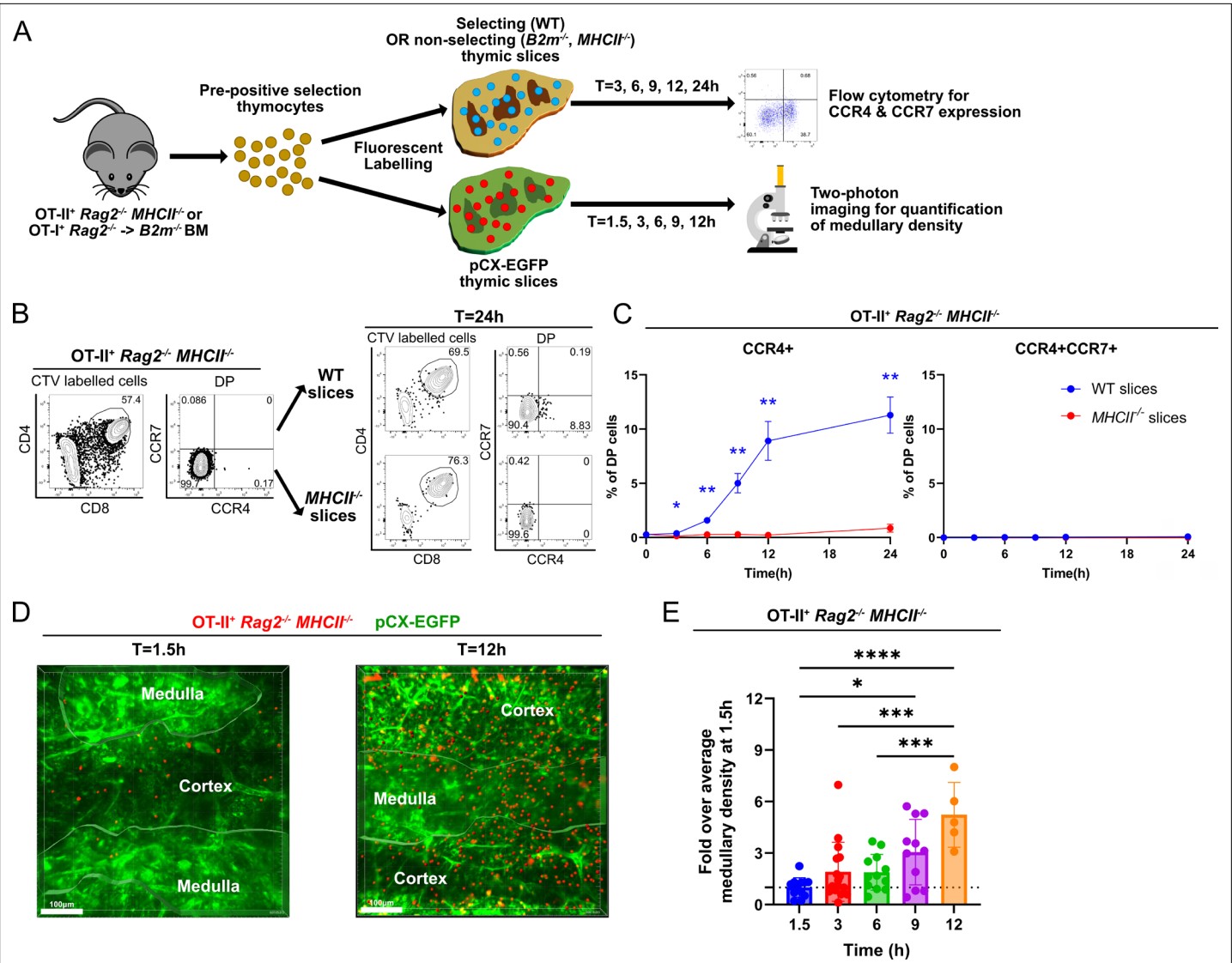

**Figure 4.** Rapid upregulation of CCR4 following positive selection of OT-II thymocytes correlates with medullary entry. (**A**) Experimental schematic of synchronized thymic slice positive selection assays to determine the timing of CCR4 and CCR7 expression by flow cytometry and medullary entry by two-photon microscopy. (**B**) Representative flow cytometry plots showing CCR4 and CCR7 expression of OT-II⁺ *Rag2⁻/⁻MHCII⁻/⁻* thymocytes added to thymic slices (left) and analyzed 24 hr after incubation in WT or *MHCII⁻/⁻* slices, as indicated (right). (**C**) Percentages of CCR4⁺ and CCR4⁺CCR7⁺ OT-II⁺ thymocytes in thymic slices of the indicated genotypes at the indicated time points. Data were compiled from three experiments with triplicate slices each. Mixed-effect analysis with Šídák's multiple comparisons correction was used for statistical analysis (mean ± SEM; *p < 0.05, **p < 0.01). (**D**) Representative maximum intensity projections of two-photon imaging data showing a pCX-EGFP thymic slice (green) containing CMTPX-labeled OT-II⁺ *Rag2⁻/⁻ MHCII⁻/⁻* thymocytes (red) at 1.5 and 12 hr after thymocytes were added to the slices. Medullary and cortical volumes were demarcated as indicated by the masked regions. Bars, 100 µm. (**E**) Quantification of medullary density of OT-II⁺ *Rag2⁻/⁻ MHCII⁻/⁻* thymocytes at indicated time points after incubation on pCX-EGFP thymic slices. Individual data points represent relative medullary input cell densities from each video compared to the average of medullary densities from all 1.5 hr videos. Data were compiled from four time-course experiments. One-way ANOVA with Tukey's multiple comparison correction was used for statistical analysis (mean ± SEM; *p < 0.05, **p < 0.01, ***p < 0.001, ****p < 0.0001).

The online version of this article includes the following figure supplement(s) for figure 4:

**Figure supplement 1.** Rapid upregulation of CCR4 following positive selection of OT-I thymocytes correlates with medullary entry.

positively selecting (wild-type [WT or pCX-EGFP]) or non-positively selecting (*MHCII⁻/⁻*) live thymic slices. At various time points, expression of CCR4 and CCR7 by OT-II thymocytes in the thymic slices was assayed by flow cytometry, or medullary entry was quantified by two-photon live-cell microscopy (*Figure 4A*).

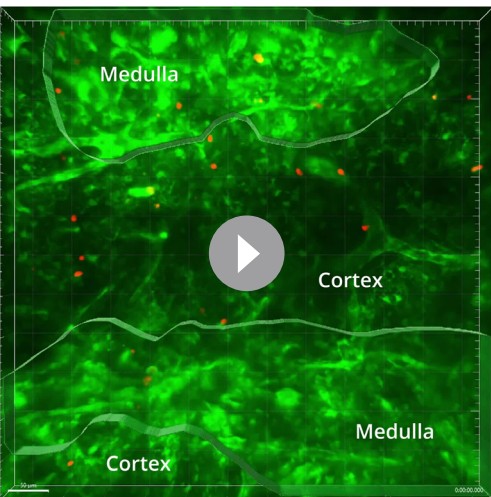

**Video 1.** Two-photon live imaging of OT-II⁺ *Rag2⁻/⁻ MHCII⁻/⁻* input thymocytes (red) migrating in a pCX-EGFP thymic slice (green) 1.5 hr after addition to positively selecting thymic slices, as shown in *Figure 3D*. Images were acquired for 15 min at 15-s time intervals through a depth of 40 µm, and maximum intensity projections are shown.

https://elifesciences.org/articles/80443/figures#video1

OT-II⁺ *Rag2⁻/⁻ MHCII⁻/⁻* DP cells upregulated CCR4 as early as 3 hr after being introduced onto positively selecting thymic slices (*Figure 4C*). The frequency of CCR4⁺ cells continued to increase over 24 hr on WT slices. Positive selection was required for initiation of CCR4 expression, as demonstrated by the failure to upregulate CCR4 on non-selecting *MHCII⁻/⁻* slices. CCR7 was not upregulated within 24 hr of initiating positive selection (*Figure 4C*). When labeled OT-II⁺ *Rag2⁻/⁻ MHCII⁻/⁻* thymocytes were introduced into positively selecting pCX-EGFP thymic slices and imaged by two-photon microscopy, they were initially localized to the cortex, consistent with our previous findings that pre-positive selection thymocytes cannot access the medulla (*Video 1*; *Ehrlich et al., 2009*). By 9–12 hr after positive selection, OT-II⁺ thymocytes entered the medulla, as evidenced by a significant increase in medullary density (*Figure 4D, E*; *Video 2*). As CCR7 is not yet expressed by these cells (*Figure 4C*), CCR4 likely drives medullary entry of early post-positive selection thymocytes.

To test if the timing of CCR4 and CCR7 upregulation and medullary entry are comparable for MHC-I-restricted thymocytes, we carried out similar experiments using pre-positive selection OT-I⁺ *Rag2⁻/⁻* thymocytes that matured in the non-selecting thymuses of *β2m⁻/⁻* hosts. Although OT-I positive selection was largely inhibited in these bone marrow chimeras, about 10% of the DP cells expressed CCR4 at baseline (*Figure 4—figure supplement 1A, B*). Nonetheless, CCR4 was upregulated as early as 3 hr after initiating positive selection in WT thymic slices, but not in non-selecting *β2m⁻/⁻* slices and increased over 24 hr (*Figure 4—figure supplement 1A, B*). CCR7 was not expressed during the first 24 hr after initiation of positive selection and only became detectable as CD8SP cells differentiated at the 72 hr time point (*Figure 4—figure supplement 1B, C*), consistent with prior observations (*Lutes et al., 2021*). Furthermore, two-photon imaging of pre-selection OT-I⁺ *Rag2⁻/⁻* thymocytes within the cortex and medulla of thymic slices showed that despite medullary entry of some thymocytes at 1.5 hr (*Video 3*), likely due to the observed baseline CCR4 expression in this model, the medullary density of OT-I thymocytes increased significantly by 12 hr after the initiation of positive selection (*Figure 4—figure supplement 1D, E*; *Video 4*).

Together, these findings show that both MHC-I- and MHC-II-restricted thymocytes upregulate CCR4 within a few hours of the initiation of positive selection and accumulate significantly in the medulla by 9–12 hr. Thymocyte medullary entry precedes CCR7 upregulation, which does not occur until 48–72 hr after positive selection. The kinetics of CCR4 upregulation are consistent with a role for CCR4 in promoting early medullary

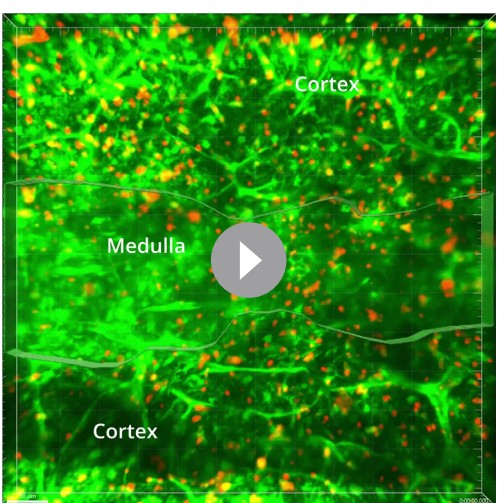

**Video 2.** Two-photon live imaging of OT-II⁺ *Rag2⁻/⁻ MHCII⁻/⁻* input thymocytes (red) migrating in a pCX-EGFP thymic slice (green) 12 hr after addition to positively selecting thymic slices, as shown in *Figure 3D*. Images were acquired for 15 min at 15-s time intervals through a depth of 40 µm, and maximum intensity projections are shown.

https://elifesciences.org/articles/80443/figures#video2

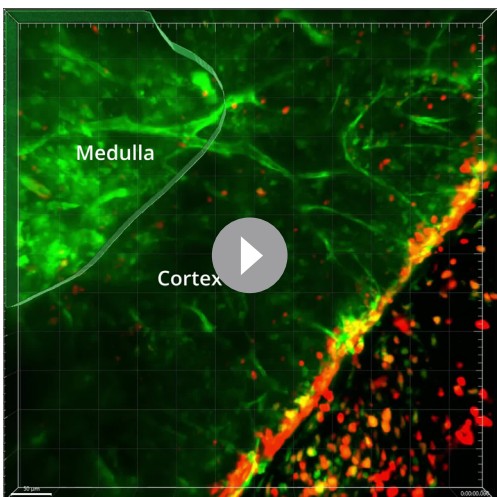

**Video 3.** Two-photon live imaging of OT-I⁺ *Rag2*⁻/⁻ input thymocytes from a *B2m*⁻/⁻ bone marrow chimera host (red) migrating in a pCX-EGFP thymic slice (green) 1.5 hr after addition to positively selecting thymic slices, as shown in *Figure 4A*. Images were acquired for 15 min at 15-s time intervals through a depth of 40 µm, and maximum intensity projections are shown.
https://elifesciences.org/articles/80443/figures#video3

entry of positively selected thymocytes. However, CCR7 is known to be required for efficient medullary entry of SP thymocytes (*Ehrlich et al., 2009*; *Kurobe et al., 2006*), raising the possibility that these two chemokine receptors promote medullary localization of different thymocyte subsets.

## CCR4 and CCR7 are required for medullary accumulation of distinct post-positive selection thymocyte subsets

To test our hypothesis that CCR4 and CCR7 are required for medullary accumulation of distinct post-positive selection thymocytes, we purified thymocyte subsets by Fluorescence Activated Cell Sorting (FACS) from WT, *Ccr4*⁻/⁻, *Ccr7*⁻/⁻, and *Ccr4*⁻/⁻; *Ccr7*⁻/⁻ (DKO) mice, labeled them with red or blue fluorescent dyes, and allowed them to migrate in pCX-EGFP live thymic slices. Two-photon time-lapse microscopy was used to image the sorted cells, so we could determine their migratory properties and densities in the medulla versus cortex (*Figure 5—figure supplement 1A*). EGFP is expressed ubiquitously in pCX-EGFP mice, such that the cortex and medulla of thymic slices can be distinguished by differences in stromal cell morphology and EGFP intensity (*Figure 5A*; *Ehrlich et al., 2009*).

Purified WT DP CD3^lo^CD69⁺ cells entered and migrated within the medulla, accumulating at a twofold higher density than in the cortex (*Figure 5A, B*; *Video 5*). As expected, the majority of the sorted early post-positive selection cells expressed CCR4, but not CCR7 (*Figure 5—figure supplement 1B, C*). Medullary enrichment of this subset was abolished when the cells were purified from *Ccr4*⁻/⁻ or DKO mice, but *Ccr7* deficiency did not diminish their medullary accumulation (*Figure 5B*). Paired analyses of WT versus *Ccr4*⁻/⁻ and *Ccr7*⁻/⁻ versus DKO DP CD3^lo^CD69⁺ cells imaged together in the same slices confirmed that *Ccr4* deficiency significantly impaired medullary accumulation of this early post-positive selection DP subset irrespective of *Ccr7* genotype, but genetic deficiency of *Ccr7* at this stage did not alter medullary accumulation (*Figure 5C*). Altogether, CCR4 is required for the twofold accumulation of DP CD3^lo^CD69⁺ cells in the medulla relative to the cortex.

Despite the fact that roughly half of the purified CD4SP SM cells expressed CCR7 (*Figure 5—figure supplement 1B, C*), which is considered to be a hallmark of thymocyte medullary entry, accumulation of these cells within the medulla did not further increase, but instead remained at a twofold medullary: cortical density ratio, comparable to that of DPCD3^lo^CD69⁺ cells (*Figure 5A, B*; *Video 6*). Medullary accumulation of this subset was significantly reduced by CCR4 deficiency, regardless of the *Ccr7* genotype (*Figure 5B, C*),

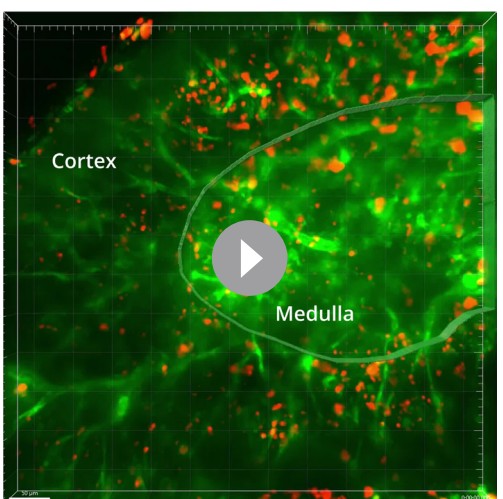

**Video 4.** Two-photon live imaging of OT-I⁺ *Rag2*⁻/⁻ input thymocytes from a *B2m*⁻/⁻ bone marrow chimera host (red) migrating in a pCX-EGFP thymic slice (green) 12 hr after addition to positively selecting thymic slices, as shown in Figure 4A. Images were acquired for 15 min at 15-s time intervals through a depth of 40 µm, and maximum intensity projections are shown.
https://elifesciences.org/articles/80443/figures#video4

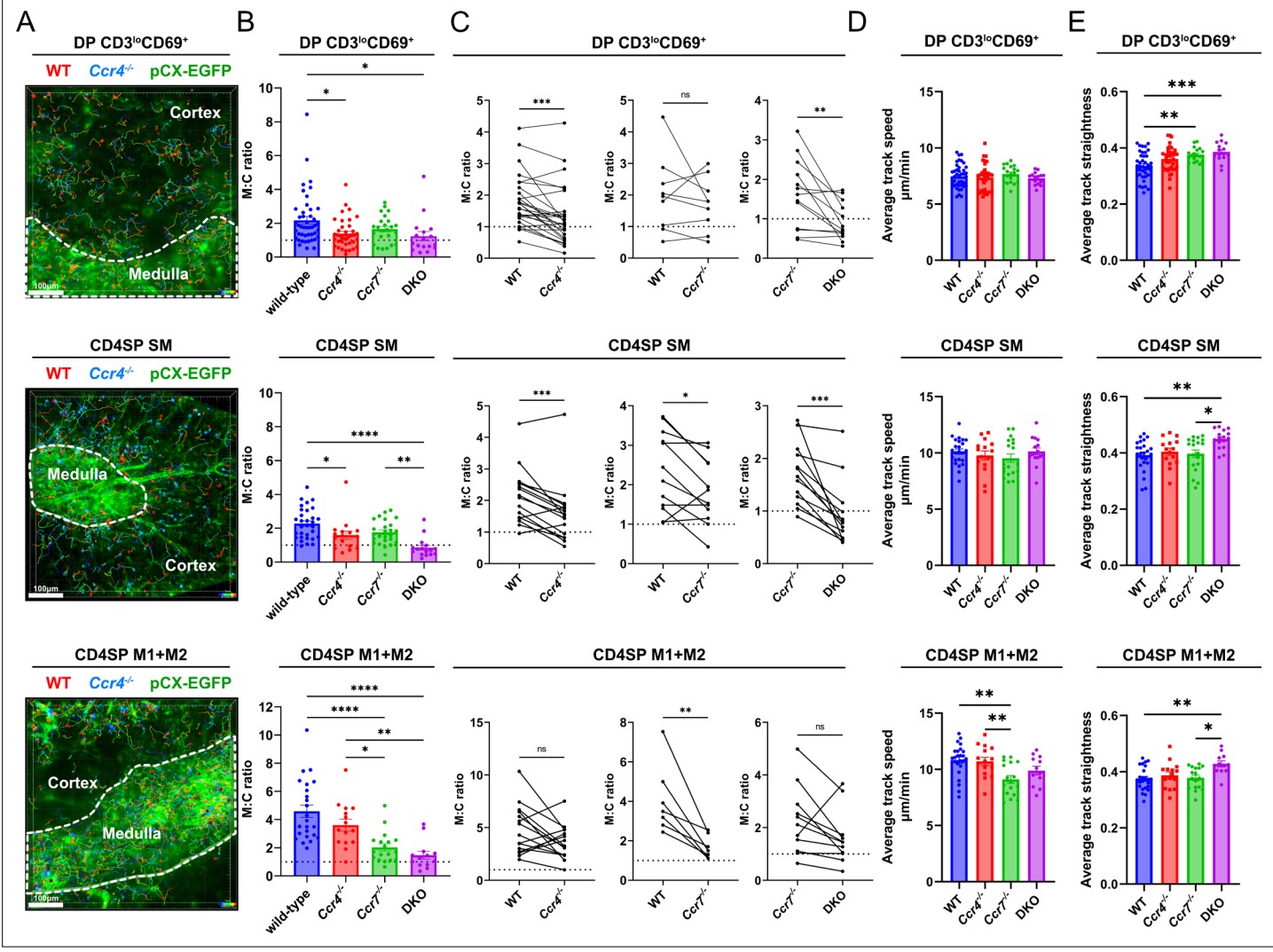

**Figure 5.** CCR4 and CCR7 are required for medullary accumulation of distinct post-positive selection thymocyte subsets. (**A**) Representative maximum intensity projections of two-photon imaging data showing pCX-EGFP thymic slices (green) containing CMTPX (red)- and Indo-1 AM (blue)-labeled FACS sorted DP CD3$^{lo}$CD69$^+$, CD4SP SM, or CD4SP M1 + M2 cells from WT or *Ccr4$^{-/-}$* mice, as indicated, imaged 1–4 hr after slice entry. The cortex and medulla are demarcated by white dotted lines, and cell tracks are color encoded for elapsed imaging time. Bars, 100 μm. (**B**) Medullary-to-cortical thymocyte density ratios were quantified from two-photon imaging data as in (**A**), with individual data points representing ratios from individual videos. (**C**) Paired analysis of medullary-to-cortical density ratios of WT versus *Ccr4$^{-/-}$*, WT versus *Ccr7$^{-/-}$*, or *Ccr7$^{-/-}$* versus DKO thymocyte subsets imaged simultaneously within the same slices, with individual data points representing ratios from single videos. Paired datasets are taken from the same data shown in (**B**), and paired *t*-tests were used for statistical analysis (**p < 0.01, ***p < 0.001). Quantification of (**D**) average thymocyte track speeds and (**E**) average track straightness from imaging data reported in (**B**). Individual data points represent average track speeds or straightness of thymocytes of the indicated genotype in individual videos. For (**B**), (**D**), and (**E**), data were compiled from 19 independent experiments. One-way ANOVA with Tukey's multiple comparisons correction was used for statistical analysis (mean ± SEM; *p < 0.05, **p < 0.01, ***p < 0.001, ****p <0.0001).

The online version of this article includes the following figure supplement(s) for figure 5:**Figure supplement 1.** Experimental approach to test if CCR4 and CCR7 are required for medullary entry of distinct FACS purified post-positive selection thymocyte subsets that differentially express CCR4 and CCR7.

indicating an important role for CCR4 in promoting medullary localization of CD4SP SM cells. In addition, paired analysis of *Ccr7$^{-/-}$* versus WT CD4SP SM cells imaged together in the same slices showed that CCR7 also contributes to medullary accumulation of CD4SP SM cells (**Figure 5C**). Together, these data indicate that CCR4 and CCR7 cooperate to promote medullary accumulation of CD4SP SM thymocytes.

CD4SP M1 + M2 thymocytes, the majority of which express only CCR7 (**Figure 5—figure supplement 1C**), accumulated in the medulla to a greater extent than either of the previous subsets, with

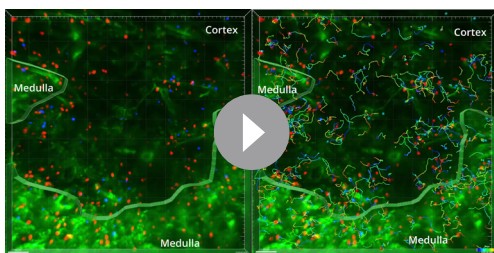

**Video 5.** Two-photon live imaging of WT (red) and *Ccr4*⁻/⁻ (blue) FACS sorted DP CD3loCD69+ thymocytes migrating in pCX-EGFP thymic slices (green), as shown in *Figure 4A*. Images were acquired for 15 min at 15-s time intervals through a depth of 40 µm, and maximum intensity projections with and without cell tracks, color coded for elapsed imaging time, are shown side by side.https://elifesciences.org/articles/80443/figures#video5

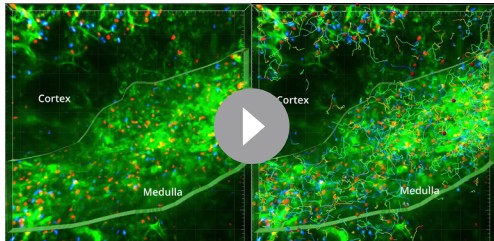

**Video 7.** Two-photon live imaging of WT (red) and *Ccr4*⁻/⁻ (blue) FACS sorted CD4SP M1 + M2 thymocyte subsets, migrating in pCX-EGFP thymic slices (green), as shown in *Figure 4A*. Images were acquired for 15 min at 15-s time intervals through a depth of 40 µm, and maximum intensity projections with and without cell tracks, color coded for elapsed imaging time, are shown side by side.

https://elifesciences.org/articles/80443/figures#video7

a ~4- to 5-fold medullary:cortical density ratio (*Figure 5B, C*; *Videos 7 and 8*), consistent with the high migration index of these cells to CCR7 ligands (*Figure 2C, D*). CCR4 deficiency did not significantly impact medullary accumulation of this subset (*Figure 5B, C*), consistent with the lack of responsiveness of M2 cells to CCR4 ligands (*Figure 2A*). However, the trend toward decreased medullary enrichment of *Ccr4*-deficient cells may reflect the activity of CCR4 in CD4SP M1 cells (*Figure 2A*). Medullary accumulation of M1 + M2 CD4SP thymocytes was highly dependent on CCR7, as demonstrated by the significant decrease in medullary accumulation of both *Ccr7*⁻/⁻ and DKO cells relative to WT or *Ccr4*⁻/⁻ cells (*Figure 5B*, *Videos 7 and 8*).

We also considered whether CCR4 and CCR7 alter thymocyte migratory properties. Deficiency in *Ccr4* and/or *Ccr7* did not impact the speed of DP CD3loCD69+ or CD4SP SM cells. However, CCR7 deficiency resulted in a significant decline in the speed of CD4SP M1 + M2 cells, consistent with our previous finding that CCR7 promotes rapid motility of CD4SP thymocytes (*Ehrlich et al., 2009*). Interestingly, double deficiency of *Ccr4* and *Ccr7* resulted in a significant increase in the average path straightness of all imaged thymocyte subsets (*Figure 5E*), suggesting that thymocytes responding to CCR4 and CCR7 ligands migrate in a more tortuous path.

Overall, CCR4 and CCR7 were required for medullary accumulation of distinct post-positive selection thymocyte subsets, largely consistent with their expression patterns and chemotactic function (*Figure 3D*). Notably, our results show that CCR4 directs early post-positive selection thymocytes into the medulla, demonstrating that lack of CCR7 expression by post-positive selection thymocytes does not preclude medullary localization. Conversely,

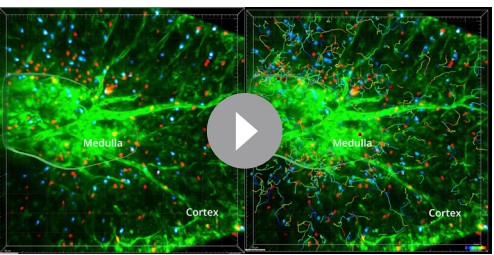

**Video 6.** Two-photon live imaging of WT (red) and *Ccr4*⁻/⁻ (blue) FACS sorted CD4SP SM thymocytes migrating in pCX-EGFP thymic slices (green), as shown in *Figure 4A*. Images were acquired for 15 min at 15-s time intervals through a depth of 40 µm, and maximum intensity projections with and without cell tracks, color coded for elapsed imaging time, are shown side by side.

https://elifesciences.org/articles/80443/figures#video6

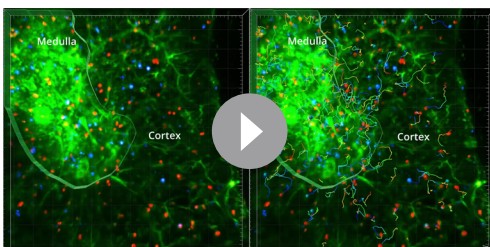

**Video 8.** Two-photon live imaging of DKO (red) and *Ccr7*⁻/⁻ (blue) FACS sorted CD4SP M1 + M2 subsets, migrating in pCX-EGFP thymic slices (green). Images were acquired for 15 min at 15-s time intervals through a depth of 40 µm, and maximum intensity projections with and without cell tracks, color coded for elapsed imaging time, are shown side by side.

https://elifesciences.org/articles/80443/figures#video8

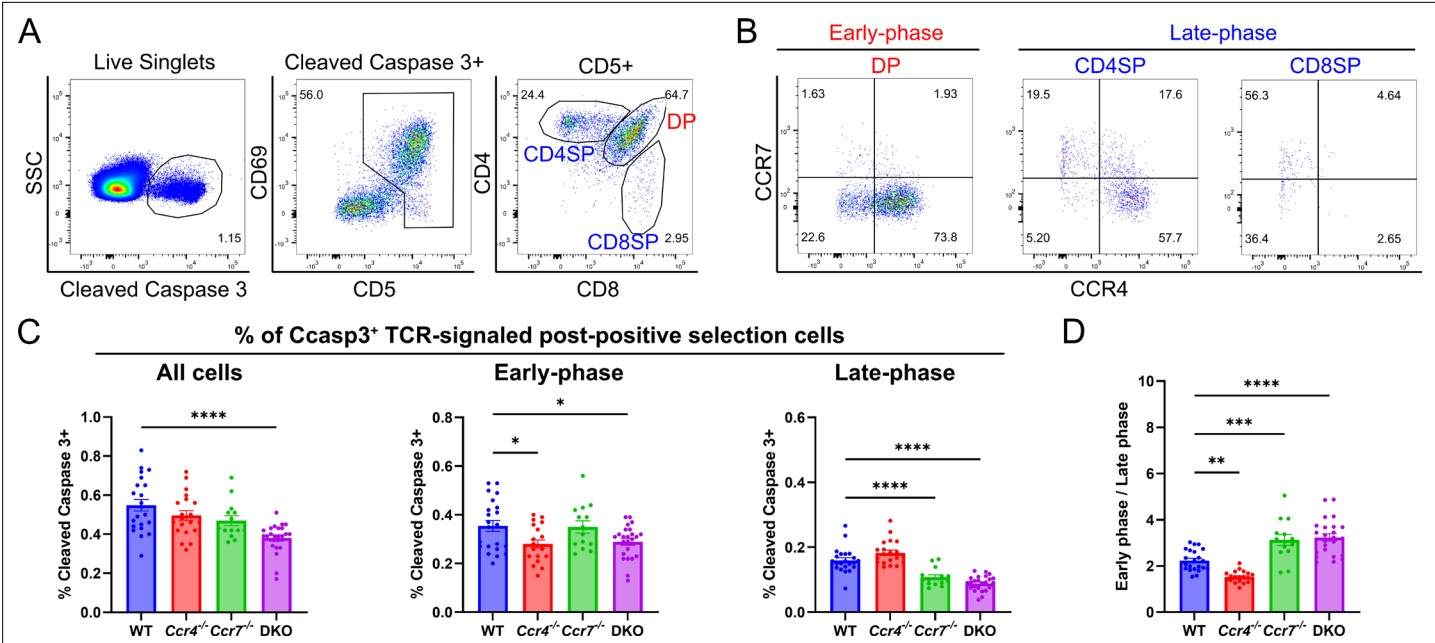

**Figure 6.** CCR4 and CCR7 contribute to early versus late phases of negative selection, respectively. (**A**) Flow cytometric gating scheme to quantify thymocytes undergoing early-phase (DP) or late-phase (CD4SP + CD8SP) negative selection. Representative data from a WT mouse are shown. (**B**) Representative flow cytometry plots showing CCR4 and CCR7 expression by thymocytes undergoing early- and late-phase negative selection, as gated in (**A**). (**C**) Frequencies of total, early-phase (DP), and late-phase (CD4SP + CD8SP) negative selection among all live thymocytes in WT, *Ccr4⁻ᐟ⁻*, *Ccr7⁻ᐟ⁻*, and DKO mice. (**D**) Ratio of thymocytes undergoing early- to late-phase negative selection in WT, *Ccr4⁻ᐟ⁻*, *Ccr7⁻ᐟ⁻*, and DKO mice. For (**C**) and (**D**), one-way ANOVA with Dunnett's multiple comparisons correction was used for statistical analysis (mean ± SEM; $N$ = 22 for WT, $N$ = 20 for *Ccr4⁻ᐟ⁻*, $N$ = 14 for *Ccr7⁻ᐟ⁻*, $N$ = 24 for DKO; *$p$ < 0.05, **$p$ < 0.01, ***$p$ < 0.001, ****$p$ < 0.0001).

The online version of this article includes the following figure supplement(s) for figure 6:

**Figure supplement 1.** Thymocyte composition and CD5 levels are not altered in *Ccr4⁻ᐟ⁻* mice.

expression of CCR7 by CD4SP SM cells does not result in enhanced medullary accumulation over the previous subset. Thus, CCR7 expression on its own is not indicative of robust medullary accumulation. Interestingly, only mature CD4SP M1 + M2 cells accumulate robustly in the medulla and migrate rapidly in a CCR7-dependent manner.

## CCR4 and CCR7 contribute to early versus late phases of negative selection, respectively

Given the differential impact of CCR4 versus CCR7 on migration and medullary accumulation of post-positive selection thymocyte subsets, we hypothesized that these chemokine receptors are required for negative selection of early versus late post-positive selection cells, respectively. Consistent with our prior observations (*Hu et al., 2015b*), the frequencies of post-positive selection thymocyte subsets were comparable between littermate control WT and *Ccr4⁻ᐟ⁻* mice (*Figure 6—figure supplement 1A*). CD5 expression levels, which are a proxy for self-reactivity (*Azzam et al., 1998*; *Hawiger et al., 2004*; *Persaud et al., 2014*), were also comparable in the absence of *Ccr4* (*Figure 6—figure supplement 1*). To test the impact of CCR4 and CCR7 on polyclonal negative selection, we quantified the frequency of cleaved-caspase 3⁺ thymocytes that had undergone TCR signaling in WT, *Ccr4⁻ᐟ⁻*, *Ccr7⁻ᐟ⁻*, and DKO mice. Intracellular cleaved-caspase 3 in TCR-signaled thymocytes has been shown to mainly reflect negative selection (*Breed et al., 2019*; *Hu et al., 2016*), although other apoptotic stimuli, such as glucocorticoid hormones, could also induce cleaved-caspase 3 in thymocytes (*Alam et al., 1997*; *Marchetti et al., 2003*). Previous studies showed that negative selection occurs in two main phases, generally defined to occur at the CCR7⁻ DP stage in the cortex versus the CCR7⁺ CD4SP stage in the medulla (*Breed et al., 2019*; *Daley et al., 2013*; *Hu et al., 2016*; *Stritesky et al., 2013*). Thus, we quantified the impact of *Ccr4* versus *Ccr7* deficiency on 'early-phase' and 'late-phase' negative selection (*Figure 6A, B*). Interestingly, the majority of early-phase DP cells undergoing negative

selection expressed only CCR4, as did ~60% of late-phase CD4SP cells; thus, the majority of thymocytes undergoing clonal deletion express CCR4, but not CCR7. CCR7 was expressed by ~40% CD4SP and 60% of CD8SP cells undergoing negative selection (*Figure 6B*). The frequency of total thymocytes undergoing negative selection declined significantly only in DKO mice (*Figure 6C*). However, CCR4 deficiency resulted in a significant decrease in the frequency of thymocytes undergoing early-phase, but not late-phase negative selection (*Figure 6C*). In contrast, CCR7 deficiency resulted in a significantly lower frequency of thymocytes undergoing late-phase negative selection, with no impact on early-phase negative selection. Double deficiency for CCR4 and CCR7 significantly diminished both early- and late-phase negative selection (*Figure 6C*). It was reported that most negative selection occurs at the earlier phase (*Breed et al., 2019*; *Stritesky et al., 2013*), which is consistent with the 2:1 ratio of WT thymocytes undergoing early- to late-phase negative selection (*Figure 6D*). This ratio falls significantly in *Ccr4*$^{-/-}$ mice and increases significantly in *Ccr7*$^{-/-}$ and DKO mice, highlighting the respective contributions of CCR4 and CCR7 to early versus late phases of negative selection, respectively (*Figure 6D*). Altogether, these data indicate that, consistent with its role in promoting medullary entry of early-post-positive selection thymocytes, CCR4 is required for efficient early-phase negative selection of polyclonal DP cells. In contrast, CCR7, which promotes medullary accumulation of more mature CD4SP and CD8SP cells, is required for efficient negative selection of late-phase CD4SP and CD8SP cells.

## CCR4 and CCR7 suppress autoinflammation in distinct tissues, with CCR4 promoting tolerance to activated APCs

Given that CCR4 and CCR7 enforced central tolerance at distinct stages of T cell development when thymocytes might interact with different APCs, these chemokine receptors could promote tolerance to distinct self-antigens. To evaluate this possibility, we assessed the presence of spontaneous autoimmunity/autoinflammation in multiple organs of WT, *Ccr4*$^{-/-}$, *Ccr7*$^{-/-}$, and DKO mice. We previously found that *Ccr4*$^{-/-}$ and *Ccr7*$^{-/-}$ mice had anti-nuclear autoantibodies by ~12 months of age (*Hu et al., 2015b*). Here, we detected anti-nuclear autoantibodies in the serum of 67% of *Ccr4*$^{-/-}$, 100% of *Ccr7*$^{-/-}$, and 75% of DKO *mice* between 5 and 6.5 months of age, relative to 22% of WT mice (*Figure 7—figure supplement 1A*). Thus, we evaluated other organs from mice in this age range for lymphocytic infiltrates or inflammation. Colon inflammation was not observed in *Ccr7*$^{-/-}$ or WT mice. Colons from *Ccr4*$^{-/-}$ mice showed mild epithelial hyperplasia, with small foci of acute inflammation. In contrast, DKO mice demonstrated moderate to severe colonic mucosal hyperplasia, with abundant acute and chronic inflammation extending into the submucosa. Active ulcerations were observed in 40% of DKO mice, with crypt abscesses and regenerative changes consistent with prior ulceration in most DKO mice (*Figure 7A*, left). Histologic scores were markedly and significantly higher in DKO mice compared to WT, with a trend toward higher scores in *Ccr4*$^{-/-}$ mice that is absent in *Ccr7*$^{-/-}$ mice (*Figure 7A*, right), suggesting that CCR4 and CCR7 collaboratively maintain self-tolerance in the colon (*Figure 7A*). Immune infiltrates were elevated in the liver, lacrimal glands, and submandibular glands of *Ccr7*$^{-/-}$ and DKO mice, but not *Ccr4*$^{-/-}$ mice at 6 months of age (*Figure 7B*, *Figure 7—figure supplement 1B–D*), indicating that CCR7 plays a dominant role in promoting tolerance to endocrine organs, consistent with the established requirement for CCR7 to mediate negative selection against mTEC-expressed TRAs, but not ubiquitous self-antigens (*Nitta et al., 2009*). Interestingly, 70–75% of *Ccr4*$^{-/-}$ mice, compared to 17–20% of WT and 0% of *Ccr7*$^{-/-}$ mice displayed moderate lymphoid hyperplasia in the spleen and mesenteric lymph nodes of 12-month-old mice (*Figure 7C*, right). Immunofluorescent staining of lymph nodes from 6-month-old *Ccr4*$^{-/-}$ mice revealed abnormal CD4 and CD8 distributions/expansions in T cell zones relative to WT mice (*Figure 7C*, left). As expected, T cell zones from *Ccr7*$^{-/-}$ mice were hypoplastic (not shown), reflecting the role of CCR7 in mediating T cell entry into lymph nodes (LNs). These results indicate that CCR4 maintains T cell homeostasis in secondary lymphoid organs.

Given the distinct contributions of CCR4 and CCR7 to establishment of central tolerance and maintenance of self-tolerance across different organs, we considered the possibility that CCR4 and CCR7 might promote thymocyte selection against antigens presented by different types of thymic APCs. Thymic APC subsets were FACS purified (*Figure 7—figure supplement 2A*), and expression of CCR4 and CCR7 ligands was quantified by qRT-PCR. Consistent with our previous study (*Hu et al., 2015b*), the CCR4 ligands CCL17 and CCL22 were expressed by Sirpα$^+$ cDC2 cells. Activated thymic CCR7$^+$

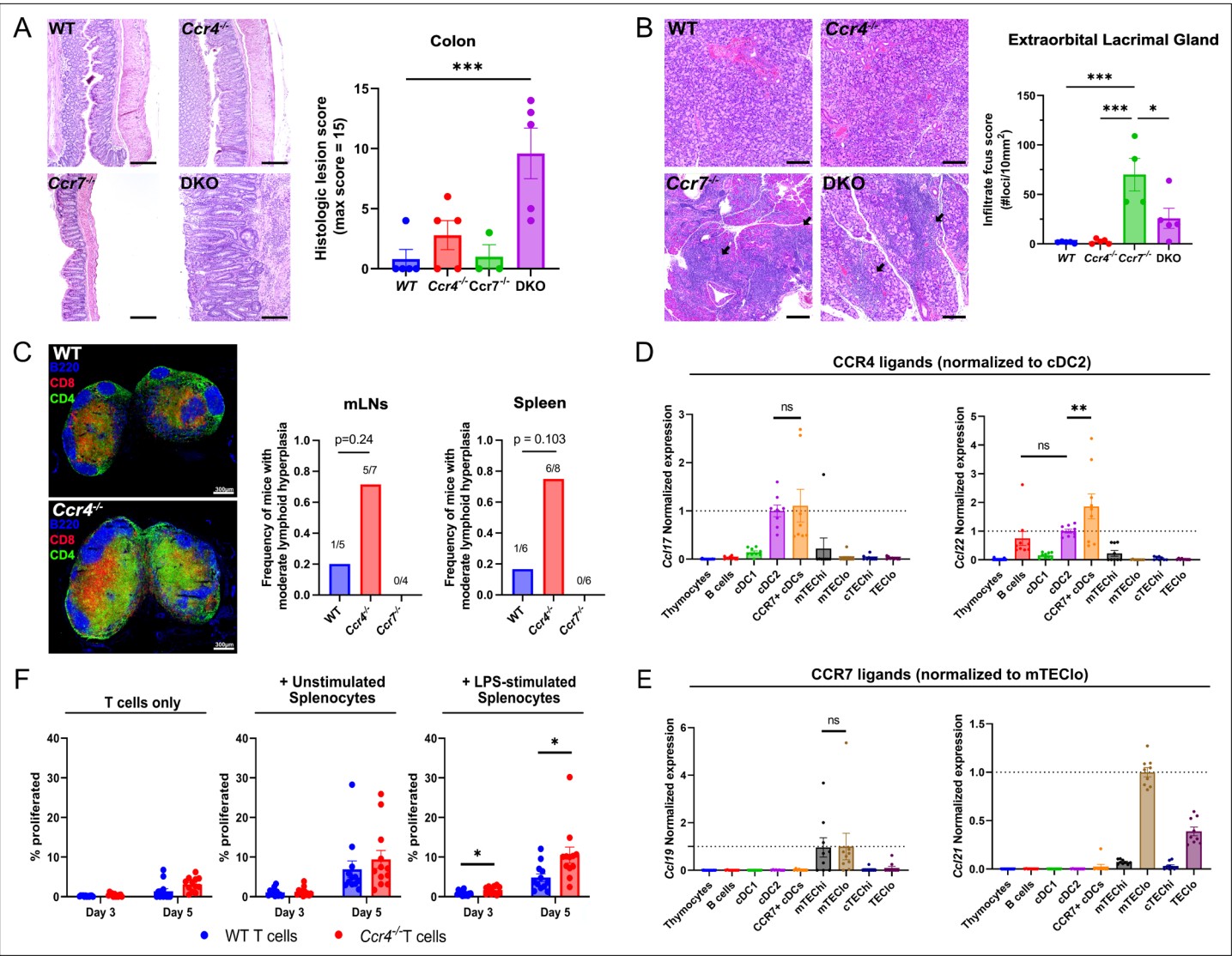

**Figure 7.** CCR4 and CCR7 promote tolerance to distinct organs, with CCR4 promoting tolerance to activated antigen-presenting cells. (**A**) Representative images of H&E stains from the colon of 5- to 6-month-old WT, *Ccr4*⁻/⁻, *Ccr7*⁻/⁻, or DKO mice (left). Bars, 200 µm. Histological lesion score was quantified by a pathologist blinded to genotype (right). One-way ANOVA was used to determine significance between groups (mean ± SEM; ***p < 0.001). (**B**) Representative images of H&E stains from extraorbital lacrimal glands of 5- to 6-month-old WT, *Ccr4*⁻/⁻, *Ccr7*⁻/⁻, or DKO mice (left). Bars, 500 µm. Arrows indicate lymphocytic infiltrate foci. The number of infiltrate foci per 10 mm² was quantified (right), and one-way ANOVA was used to test significance between groups (mean ± SEM; *p < 0.05, ***p > 0.001). (**C**) Representative immunofluorescent images of CD4 (green), CD8 (red), and B220 (blue) on cryosections of inguinal lymph nodes from 5- to 6-month-old WT and *Ccr4*⁻/⁻ mice (left). Frequency of moderate lymphoid hyperplasia in mLNs and spleens from 12- to 13-month-old WT, *Ccr4*⁻/⁻, and *Ccr7*⁻/⁻ mice, as determined by a veterinary pathologist, blinded to genotype (right). Fisher's exact test was used to determine significance. Bars, 300µm. Expression of CCR4 ligands (**D**) and CCR7 ligands (**E**) by distinct thymic antigen-presenting cell subsets, assessed by quantitative reverse-transcription polymerase chain reaction (qRT-PCR). Expression levels were normalized to subsets previously reported to express the respective ligands: CCL17 and CCL22 expression were normalized to cDC2 (*Hu et al., 2015b*), and CCL19 and CCL21 expression to mTEC^lo (*Ki et al., 2014*; *Misslitz et al., 2004*). Data were compiled from three independent experiments with triplicates. One-way ANOVA with Dunnett's multiple comparisons correction was used for statistical analysis (mean ± SEM; ns: p > 0.05, **p < 0.01). (**F**) The percentage of WT or *Ccr4*⁻/⁻ T cells that proliferated at days 3 and 5 when cultured without splenocytes, with unstimulated splenocytes, or with lipopolysaccharide (LPS)-stimulated splenocytes, as indicated. Data were compiled from four independent experiments with triplicate wells, and unpaired *t*-tests with Holm–Šídák's multiple comparisons correction was used for statistical analysis (mean ± SEM; *p < 0.05).

The online version of this article includes the following figure supplement(s) for figure 7:

**Figure supplement 1.** CCR4 and CCR7 support self-tolerance in different organs.

**Figure supplement 2.** CCR4 promotes tolerance to activated antigen-presenting cells (APCs).

cDCs (*Ardouin et al., 2016*; *Hu et al., 2017*; *Oh et al., 2018*) also expressed CCR4 ligands, with the highest levels of CCL22 expressed by activated DCs (*Figure 7D*). Thymic B cells also expressed CCL22, consistent with a recent report that CCL22 expressed by activated B cells promotes interactions with T cells in the germinal center (*Liu et al., 2021*). The CCR7 ligands CCL19 and CCL21 were expressed by mTEC subsets, with the highest levels of CCL21 expression by mTEC$^{lo}$ cells (*Figure 7E*), consistent with previous reports (*Bornstein et al., 2018*; *Kozai et al., 2017*). These results suggest that CCR4 may promote thymocyte interactions with B cells and cDCs, including activated CCR7$^+$ cDCs that express elevated levels of MHC-II and co-stimulatory molecules (*Ardouin et al., 2016*; *Hu et al., 2017*), while CCR7 may promote thymocyte interactions primarily with mTECs.

Because a large proportion of thymic B cells (*Cepeda et al., 2018*) and cDCs (*Ardouin et al., 2016*; *Hu et al., 2017*) have activated phenotypes, and T cell zones in the lymph nodes of *Ccr4*$^{-/-}$ mice are hyperplastic (*Figure 7C*), we considered the possibility that CCR4-directed negative selection could facilitate establishment of central tolerance to antigens expressed by activated APCs. To explore this hypothesis, CD45.1$^+$ congenic splenocytes were stimulated overnight with toll-like receptor (TLR) ligands. We confirmed that stimulation with TLR ligands induced activation of the splenic cDCs and B cells based on upregulation of MHC-II, CD80, and CD86 (*Figure 7—figure supplement 2B*). The activated splenocytes were then co-cultured with CD45.2$^+$ CD4$^+$ conventional T cells isolated from WT or *Ccr4*$^{-/-}$ spleens, and the frequency of T cells induced to proliferate was quantified after 3–5 days (*Figure 7F*). Notably, LPS-stimulated splenocytes induced more proliferation of *Ccr4*$^{-/-}$ versus WT CD4$^+$ T cells at days 3 and 5 (*Figure 7G*). Splenocytes that were unstimulated or activated by other TLR ligands did not preferentially induce *Ccr4*$^{-/-}$ T cell proliferation (*Figure 7E*; *Figure 7—figure supplement 2C*). Taken together, these data suggest that CCR4 promotes thymocyte central tolerance to self-antigens expressed by LPS-activated APCs, possibly by promoting interactions with such APCs in the thymus.

## Discussion

Our data demonstrate that CCR4 and CCR7 play distinct roles in thymocyte localization and the induction of T cell central tolerance. We find that CCR4 is upregulated by thymocytes within a few hours of the initiation of positive selection, largely coincident with the timing of medullary entry. CCR4 is expressed by almost all early post-positive selection DP CD3$^{lo}$CD69$^+$ cells and is required for their twofold accumulation in the medulla and efficient negative selection. Notably, the majority of thymocytes undergoing negative selection are DP CD3$^{lo}$CD69$^+$ cells, which do not express CCR7. Thus, while the role of CCR4 in negative selection has been questioned (*Cowan et al., 2014*), our analyses reveal an important role for this chemokine receptor in localization and central tolerance of early post-positive selection polyclonal thymocytes, in keeping with medullary localization of the CCR4 ligand CCL22. In contrast, our study and a previous report (*Lutes et al., 2021*) show that CCR7 is not upregulated until 48–72 hr after positive selection. CCR7 is expressed by about half of the CD4SP SM cells, which continue to express CCR4 and respond to CCR4 ligands. In keeping with dual expression, CCR4 and CCR7 cooperate to induce the twofold accumulation of CD4SP SM cells in the medulla. As thymocytes mature further to the CD4SP and CD8SP M1 and M2 stages, CCR7 is upregulated, CCR4 is downregulated, and the cells become increasingly chemotactic to CCR7 ligands. We find that CCR7 is required for the robust accumulation of CD4SP M1 + M2 cells in the medulla and for late-phase negative selection of CD4SP and CD8SP thymocytes. The finding that the overall frequency of negative selection in the thymus is reduced significantly only in the absence of both CCR4 and CCR7 further indicates that these chemokine receptors both contribute to the induction of central tolerance. Altogether, our findings show a temporally correlated shift after positive selection from a requirement for CCR4 to a requirement for CCR7 for thymocyte medullary localization and negative selection.

Several of our findings raise questions about the current model of negative selection, which assumes that CCR7 expression is a surrogate indicator of thymocyte medullary versus cortical localization (*Breed et al., 2019*; *Hu et al., 2016*). This assumption is based on studies, including our own, showing that CCR7 is expressed by CD4SP and CD8SP thymocytes, which are localized mainly within the thymic medulla at steady state, as evidenced by immunostaining (*Ehrlich, 2016*). In addition, prior studies showed that accumulation of CD4SP cells in the medulla is CCR7 dependent (*Ehrlich et al., 2009*; *Ueno et al., 2004*). Consistent with these findings, we confirm that robust medullary accumulation of the more mature CD4SP M1 and M2 cells is CCR7 dependent. Equating CCR7 expression with

medullary localization has led to a model in which the early phase of negative selection is thought to impact CCR7⁻ DP cells in the cortex, while the later phase impacts more mature CCR7⁺ cells in the medulla (*Breed et al., 2019*; *Hu et al., 2016*). In support of cortical negative selection, apoptotic TCR transgenic cells were located in the cortex of the H-Y^CD4 model of negative selection (*McCaughtry et al., 2008*). Also, TCR-signaled thymocytes are present at the cortical side of the CMJ in *Bim*⁻/⁻ mice, in which apoptosis following strong TCR signaling is impaired (*Stritesky et al., 2013*), suggesting thymocytes encounter negatively selecting self-antigens in the cortex. Furthermore, CCR7 deficiency impairs negative selection to TRAs, but not to ubiquitous self-antigens (*Kurobe et al., 2006*; *Nitta et al., 2009*), indicating that CCR7⁺ mature thymocytes undergo negative selection in the medulla, where TRAs are expressed by mTECs.

While our data confirm that CCR7 is critical for medullary accumulation of more mature thymocyte subsets, we find that CCR7 expression is not a reliable indicator of medullary localization in earlier post-positive selection thymocytes. CCR7⁻ DP CD3^lo^CD69⁺ enter the medulla, where they accumulate in a CCR4-dependent manner with a twofold higher density than in the cortex, so CCR7 is not required for medullary entry. Conversely, although almost half of CD4SP SM cells express CCR7, their medullary: cortical density is also twofold. CCR7 deficiency only modestly diminished medullary accumulation of CD4SP SM cells, consistent with their moderate chemotaxis toward CCR7 ligands; instead, CCR4 played a greater role in medullary accumulation of this subset. Thus, the two earliest post-positive selection subsets accumulate in the medulla at similar densities, despite CCR7 upregulation by one of these subsets. DP CD3^lo^CD69⁺ and CD4SP SM cells both migrate in the cortex and in the medulla and could thus encounter APCs presenting cognate self-antigens that drive negative selection in either region. Based on the twofold enrichment of DP CD3^lo^CD69⁺ cells in the medulla, a substantial proportion of CCR7⁻ early-phase negative selection may occur in the medulla. Moreover, some late-phase negative selection, defined by either CCR7 expression or maturation to the CD4SP SM stage, may occur in the cortex. These data indicate that CCR7 expression cannot be used as a reliable surrogate biomarker of medullary localization. The location in which thymocytes undergo negative selection impacts which APC subsets will be encountered, altering the spectrum of self-antigens to which thymocytes are tolerized. Thus, it will be important to determine where distinct thymocyte subsets undergo negative selection and which APCs promote tolerance of these different subsets.

It has recently been suggested that CXCR4 expression must be extinguished after positive selection to release thymocytes from binding to cTECs in the cortex, thus enabling medullary entry (*Kadakia et al., 2019*). However, we found that while CXCR4 is partially downregulated on the cell surface of post-positive selection thymocytes, all subsets except CD4SP SM and DP CD3⁺ CD69⁺ cells undergo chemotaxis to CXCL12, despite their accumulation in the medulla. These findings indicate that CXCR4 expression and activity are compatible with medullary localization. This seeming discrepancy may reflect the overexpression system used to reveal that persistent CXCR4 expression prevented thymocyte medullary entry, perhaps resulting in superphysiological tethering of cells to the cortex (*Kadakia et al., 2019*). In keeping with the ability of CXCR4⁺ thymocytes to enter the medulla, higher CXCR4 surface expression and responsiveness are observed in mature CD4SP and CD8SP M1 and M2 subsets, which accumulate in the medulla to the greatest extent. It is notable that chemotactic activity of CXCR4 is completely extinguished at the CD4SP SM stage, despite continued CXCR4 expression, but not at the more mature CD4SP and CD8SP stages, suggesting a thymocyte-intrinsic mechanism of diminished responsiveness to CXCR4, similar to the lack of responsiveness to CCR7 ligands by this subset. It is possible that the lack of CXCR4 activity in CD4SP SM cells may permit medullary entry, consistent with the need to release post-positive selection thymocytes from binding to cTECs (*Kadakia et al., 2019*); however, DP CD3^lo^CD69⁺ cells enter the medulla just as efficiently as CD4SP SM cells despite continued CXCR4 responsiveness, suggesting factors other than reduced CXCR4 activity likely enable medullary entry. The role for increased CXCR4 activity in more mature SP thymocytes remains to be determined but could impact thymocyte egress. Identifying intrinsic regulators of chemokine receptor responsiveness in different thymocyte subsets will further our understanding of the complex mechanisms underlying the orchestration of thymocyte movement and selection in the thymus.

A major outstanding question raised by our findings is whether the combinatorial and temporally regulated changes in chemokine receptor expression on post-positive selection thymocytes drive interactions with distinct APC subsets. Here we show that CCR7 ligands are expressed by mTECs,

while CCR4 ligands are expressed mainly by cDC2s and activated cDCs, with some expression by thymic B cells. Thus, CCR7 activity may drive interactions with mTECs, possibly explaining why CCR7 is particularly required for negative selection to TRAs, consistent with the finding that CCR7 deficiency results in autoimmune infiltrates in endocrine organs, as occurs in Aire-deficient mice (*Anderson et al., 2002*; *Kurobe et al., 2006*; *Nitta et al., 2009*). In contrast, CCR4 activity may promote interactions with and tolerance to cDC2s, B cells, and activated cDCs, consistent with our finding that $Ccr4^{-/-}$ mice showed T cell hyperplasia in secondary lymphoid organs and that $Ccr4^{-/-}$ T cells undergo increased proliferation to LPS-activated splenic APCs. Recently, a subset of cDC2s has been shown to present circulating antigens to induce negative selection, and another subset has been shown to traffic microbial antigens to the thymus, impacting thymocyte selection (*Atibalentja et al., 2009*; *Vollmann et al., 2021*; *Zegarra-Ruiz et al., 2021*). B cells also present self-antigens to induce thymocyte negative selection (*Perera et al., 2016*; *Yamano et al., 2015*). While we find CCL22 is expressed predominantly in the medulla, and CCR4 promotes medullary accumulation of early-post-positive selection DPs, our study does not rule out the possibility that CCR4 could also enforce interactions between CCR4-responsive thymocytes and cortical DCs expressing CCL22, which could contribute to cortical negative selection. This possibility is particularly intriguing in light of the fact that DP CD3$^{lo}$CD69$^+$ and CD4SP SM cells respond to CCR4 ligands and migrate both in the cortex and medulla. The finding that CCR4 and CCR7 cooperate to prevent inflammation in the colon is unexpected and warrants further exploration. Whether the inflammatory phenotypes observed reflect impaired central tolerance due to diminished interactions between thymocytes and distinct thymic APCs and/or impaired peripheral tolerance remains to be determined.

Altogether, our findings suggest a layered process of central-tolerance induction in which CCR4 first promotes interactions with DCs and B cells, driving self-tolerance to activated APCs, which may be present in both the cortex and/or medulla. As thymocytes mature further, CCR7 is expressed and becomes active, drawing the cells robustly into the medulla, where the partially tolerized repertoire can be focused on tolerance induction to the sparse TRAs. One limitation of our current system is that ex vivo live thymic slices are not equivalent to in situ thymi, as they lack circulation, for example. We note, however, that short-term live thymic slice cultures have been widely used to investigate the development, localization, migration, and positive and negative selection of thymocytes, as they have been shown to faithfully reflect these in vivo processes, including confirming that CCR7 signaling induces chemotaxis of mature thymocytes from the cortex into the medulla (*Au-Yeung et al., 2014*; *Dzhagalov et al., 2013*; *Ehrlich et al., 2009*; *Lancaster et al., 2019*; *Melichar et al., 2013*; *Ross et al., 2014*). Because our study measured differences in medullary accumulation of thymocytes that differed genetically only in expression of CCR4 and/or CCR7, we attribute altered localization to the impact of these chemokine receptors on thymocyte migration. However, we note the possibility that CCR4 and CCR7 could signal in conjunction with survival cues, like cytokines, localized to different thymic microenvironments, which could also impact survival of thymocytes in different regions. As thymocytes were imaged within a few hours of entering slices, differential survival in the cortex and medulla due to cytokine cues is somewhat less likely than differential migration. Multiple aspects of our revised model of thymocyte migration and central tolerance, including the impact of CCR4 and CCR7 on interactions with distinct APCs during tolerance induction and on selection of distinct TCR clones will be tested in future studies.

## Materials and methods

**Key resources table**

| Reagent type (species) or resource | Designation | Source or reference | Identifiers | Additional information |
|---|---|---|---|---|
| gene (*Mus musculus*) | Ccr4 | GenBank | Gene ID: 12773 | |
| gene (*Mus musculus*) | Ccr7 | GenBank | Gene ID: 12775 | |
| strain, strain background (*Mus musculus*) | C57BL6/J | Jackson Laboratory | Strain #: 000664 | |
| genetic reagent (*Mus musculus*) | B6.SJL-Ptprc$^a$ PepC$^b$/BoyJ (CD45.1) | Jackson Laboratory | Strain #: 002014 | |

*Continued on next page*

*Continued*

| Reagent type (species) or resource | Designation | Source or reference | Identifiers | Additional information |
|---|---|---|---|---|
| genetic reagent (*Mus musculus*) | C57BL/6-Tg(TcraTcrb)1100Mjb/J (OT-I) | Jackson Laboratory | Strain #: 003831 | |
| genetic reagent (*Mus musculus*) | B6.Cg-Tg(TcraTcrb)425Cbn/J (OT-II) | Jackson Laboratory | Strain #: 004194 | |
| genetic reagent (*Mus musculus*) | B6(Cg)-Rag2$^{tm1.1Cgn}$/J (Rag2$^{-/-}$) | Jackson Laboratory | Strain #: 008449 | |
| genetic reagent (*Mus musculus*) | B6.129P2(C)Ccr7$^{tm1Rfo}$r/J (Ccr7$^{-/-}$) | Jackson Laboratory | Strain #: 006621 | |
| genetic reagent (*Mus musculus*) | B6.129S2-H2$^{dlAb1-Ea}$/J (H2$^{-/-}$) | Jackson Laboratory | Strain #: 003584 | |
| genetic reagent (*Mus musculus*) | B6.129P2-B2m$^{tm1Unc}$/DcrJ (B2m$^{-/-}$) | Jackson Laboratory | Strain #: 002087 | |
| genetic reagent (*Mus musculus*) | C57BL/6-Tg(Nr4a1-EGFP/cre)820Khog/J (Nr4a1$^{GFP}$) | Jackson Laboratory | Strain #: 016617 | |
| genetic reagent (*Mus musculus*) | Ccr4$^{-/-}$ | *Chvatchko et al., 2000* | | Generously provided by A.D. Luster (Massachusetts General Hospital, Boston, MA) |
| genetic reagent (*Mus musculus*) | pCX-EGFP | *Wright et al., 2001* | | Generously provided by Irving L. Weissman (Stanford University, Stanford, CA) |
| genetic reagent (*Mus musculus*) | Rag2p-GFP | *Boursalian et al., 2004* | | Generously provided by Ellen R. Richie (the University of Texas MD Anderson Center, Houston, TX) |
| genetic reagent (*Mus musculus*) | Ccr4$^{-/-}$; Ccr7$^{-/-}$(DKO) | This paper | | See Materials and Methods, Section "Mice" |
| genetic reagent (*Mus musculus*) | OT-I Rag2$^{-/-}$ | This paper | | See Materials and Methods, Section "Mice" |
| genetic reagent (*Mus musculus*) | OT-II Rag2$^{-/-}$ H2$^{-/-}$ | This paper | | See Materials and Methods, Section "Mice" |
| antibody | PE/Cyanine 7 Anti-mouse CD3, Rat Monoclonal, clone 17 A2 | BioLegend | Cat # 100220; RRID:AB_1732057 | (1:200) |
| antibody | PE Anti-mouse CCR4, Armenian Hamster Monoclonal, clone 2 G12 | BioLegend | Cat # 131204; RRID:AB_1236367 | (1:200) |
| antibody | APC Anti-mouse CCR7, Rat Monoclonal, clone 4B12 | BioLegend | Cat # 120108; RRID:AB_389234 | (1:200) |
| antibody | Brilliant Violet 510 Anti-mouse CD4, Rat Monoclonal, clone RM4-5 | BioLegend | Cat # 100559; RRID:AB_2562608 | (1:200) |
| antibody | PE/Cyanine7 Anti-mouse CD5, Rat Monoclonal, clone 53–7.3 | BioLegend | Cat # 100622; RRID:AB_2562773 | (1:200) |
| antibody | FITC Anti-mouse CD8a, Rat Monoclonal, clone 53–6.7 | BioLegend | Cat # 100706; RRID:AB_312745 | (1:200) |
| antibody | Alexa Flour 700 Anti-mouse CD11b, Rat Monoclonal, clone M1/70 | BioLegend | Cat # 101222; RRID:AB_493705 | (1:200) |
| antibody | Pacific Blue Anti-mouse CD11c, Armenian Hamster Monoclonal, clone N418 | BioLegend | Cat # 117322; RRID:AB_755988 | (1:200) |
| antibody | PerCP/Cyanine5.5 Anti-mouse CD19, Rat Monoclonal, clone 1D3/CD19 | BioLegend | Cat # 152405; RRID:AB_2629814 | (1:200) |
| antibody | APC Anti-mouse CD25, Rat Monoclonal, clone PC61 | BioLegend | Cat # 102012; RRID:AB_312861 | (1:200) |
| antibody | APC Anti-mouse CD45.1, Mouse (A. SW) monoclonal, clone A20 | BioLegend | Cat # 110714; RRID:AB_313503 | (1:200) |
| antibody | PE Anti-mouse CD45.2, Mouse (SJL) monoclonal, clone 104 | BioLegend | Cat # 109808; RRID:AB_313445 | (1:200) |

*Continued on next page*

*Continued*

| Reagent type (species) or resource | Designation | Source or reference | Identifiers | Additional information |
|---|---|---|---|---|
| antibody | Biotin Anti-mouse CD69, Armenian Hamster monoclonal, clone H1.2F3 | BioLegend | Cat # 104504; RRID:AB_313107 | (1:200) |
| antibody | Brilliant Violet 510 Anti-mouse CD90.2, Rat monoclonal, clone 30 H12 | BioLegend | Cat # 105335; RRID:AB_2566587 | (1:200) |
| antibody | PE/Dazzle 594 Anti-mouse CXCR4, Rat monoclonal, clone L276F12 | BioLegend | Cat # 146514; RRID:AB_2563683 | (1:200) |
| antibody | Alexa Flour 488 Anti-mouse Cleaved Caspase 3 (Asp175), Rabbit polyclonal | Cell Signaling Technology | Cat # 9669 S | (1:200) |
| antibody | PE Anti-mouse EpCAM, Rat monoclonal, clone G8.8 | BioLegend | Cat # 118206; RRID:AB_1134172 | (1:200) |
| antibody | APC/Cyanine7 Anti-mouse F4/80, Rat monoclonal, clone BM8 | BioLegend | Cat # 123118; RRID:AB_893477 | (1:200) |
| antibody | PE/Cyanine5 Anti-mouse Gr1, Rat monoclonal, clone RB6-8C5 | BioLegend | Cat # 108410; RRID:AB_313375 | (1:200) |
| antibody | PerCP/Cyanine5.5 Anti-mouse H-2Kb (MHCI), Mouse (BALB/c) monoclonal, clone AF6-88.5 | BioLegend | Cat # 116516; RRID:AB_1967133 | (1:200) |
| antibody | PE/Cyanine7 Anti-mouse I-A/I-E (MHCII), Rat Monoclonal, clone M5/114.15.2 | BioLegend | Cat # 107630; RRID:AB_2069376 | (1:200) |
| antibody | Alexa Flour 488 Anti-mouse PDCA1, Rat Monoclonal, clone 927 | BioLegend | Cat # 127012; RRID:AB_1953287 | (1:200) |
| antibody | PE/Cyanine5 Anti-mouse Ter-119, Rat monoclonal, clone TER-119 | BioLegend | Cat # 116209; RRID:AB_313710 | (1:200) |
| antibody | APC Anti-mouse Sirpα, Rat monoclonal, clone P84 | BioLegend | Cat # 144014; RRID:AB_2564061 | (1:200) |
| antibody | Brilliant Violet 650Anti-mouse XCR1, mouse monoclonal, clone ZET | BioLegend | Cat # 148220; RRID:AB_2566410 | (1:200) |
| antibody | Biotin Anti-mouse CD11c, Armenian Hamster monoclonal, clone N418 | BioLegend | Cat # 117303; RRID:AB_313772 | (1:50) |
| antibody | APC Anti-mouse CD4, Rat monoclonal, clone GK1.5 | BioLegend | Cat # 100412; RRID:AB_312696 | (1:100) |
| antibody | APC Anti-mouse CD31, Rat monoclonal, clone 390 | BioLegend | Cat# 102410; RRID:AB_312905 | (1:100) |
| antibody | Biotin anti-mouse/human CD45R/B220, Rat monoclonal, clone RA3-6B2 | BioLegend | Cat # 103204; RRID:AB_312989 | (1:200) |
| antibody | Purified Rat anti-mouse B220, Rat monoclonal, clone RA3.3A1/6.1 | BioXCell | Cat # BE0067 | (1:100) |
| antibody | Purified Rat anti-mouse CD3, Rat monoclonal, clone 17 A2 | BioXCell | Cat # BE0002 | (1:50) |
| antibody | Purified Rat anti-mouse CD8, Rat monoclonal, clone 53.6.72 | BioXCell | Cat # BE0004-1 | (1:100) |
| antibody | Purified Rat anti-mouse CD25, Rat monoclonal, clone PC-61.5.3 | BioXCell | Cat # BE0012 | (1:100) |
| antibody | Purified Rat anti-mouse CD11b, Rat monoclonal, clone M1/70 | BioXCell | Cat # BE0007 | (1:100) |
| antibody | Purified Rat anti-mouse Gr-1, Rat monoclonal, clone RB6-8C5 | BioXCell | Cat # BE0075 | (1:100) |
| antibody | Purified Rat anti-mouse Ter-119, Rat monoclonal, clone TER-119 | BioXCell | Cat # BE0183 | (1:100) |
| antibody | Alexa Fluor 594 AffiniPure Donkey anti-mouse IgG, Donkey polyclonal | Jackson ImmunoResearch | Cat # 715585151; RRID:AB_2340855 | (1:100) |

*Continued on next page*

*Continued*

| Reagent type (species) or resource | Designation | Source or reference | Identifiers | Additional information |
|---|---|---|---|---|
| antibody | Goat anti-mouse CCL22, Polyclonal, Goat polyclonal | R&D Systems | Cat # AF439; RRID:AB_355360 | (1.2 ug/mL) |
| antibody | Normal Goat IgG control, Goat polyclonal | R&D Systems | Cat # AB-108-C; RRID:AB_354267 | (1:200) |
| antibody | CD8a- Alexa Fluor 594, rat monoclonal, clone 53.6.7 | BioLegend | Cat# 100758 | (1:100) |
| Other flow cytometry reagents | Qdot 605 Streptavidin Conjugate | Invitrogen | Cat # Q10101MP | (1:200) |
| Other immunostaining reagents | Streptavidin-Alexa Fluor 488 | Invitrogen | Cat# S32354 | |
| Other flow cytometry reagents | Ulex Europaeus Agglutinin I (UEAI), Biotinylated | Vector Laboratories | SKU B-1065–2 | (1:1000) |
| sequence-based reagent | CCL17 Forward | *Cowan et al., 2014* | qPCR Primer | 5'-AGTGGAGTGTTCCAGGGATG-3' |
| sequence-based reagent | CCL17 Reverse | *Cowan et al., 2014* | qPCR Primer | 5'-CCAATCTGATGGCCTTCTTC-3' |
| sequence-based reagent | CCL19 Forward | *Kurd and Robey, 2016* | qPCR Primer | 5'-GCTAATGATGCGGAAGACTG-3' |
| sequence-based reagent | CCL19 Reverse | *Kurd and Robey, 2016* | qPCR Primer | 5'-ACTCACATCGACTCTCTAGG-3' |
| sequence-based reagent | CCL21 Forward | *Seach et al., 2008* | qPCR Primer | 5'-GCAGTGATGGAGGGGGTCAG-3' |
| sequence-based reagent | CCL21 Reverse | *Seach et al., 2008* | qPCR Primer | 5'-CGGGGGTGAGAACAGGATTGC-3' |
| sequence-based reagent | CCL22 Forward | *Hu et al., 2015b* | qPCR Primer | 5'-AGGTCCCTATGGTGCCAATGT-3' |
| sequence-based reagent | CCL22 Reverse | *Hu et al., 2015b* | qPCR Primer | 5'- CGGCAGGATTTTGAGGTCCA-3' |
| sequence-based reagent | β-actin Forward | *Camara et al., 2019* | qPCR Primer | 5'- CACTGTCGAGTCGCGTCCA-3' |
| sequence-based reagent | β-actin Reverse | *Camara et al., 2019* | qPCR Primer | 5'- CATCCATGGCGAACTGGTGG-3' |
| peptide, recombinant protein | Recombinant Murine CCL17 | Peprotech | Cat # 250–43 | |
| peptide, recombinant protein | Recombinant Murine CCL19 | Peprotech | Cat # 250-27B | |
| peptide, recombinant protein | Recombinant Murine CCL21 | Peprotech | Cat # 250–13 | |
| peptide, recombinant protein | Recombinant Murine CCL22 | Peprotech | Cat # 250–23 | |
| peptide, recombinant protein | Recombinant Murine CXCL12 | Peprotech | Cat # 250-20B | |
| peptide, recombinant protein | Recombinant Mouse CCL25 | R&D Systems | Cat # 481-TK-025 | |
| commercial assay or kit | Fixation / Permeablization Kit (RUO) | BD | Cat # 554714 | |
| commercial assay or kit | CellTrace Violet cell proliferation kit | Invitrogen | Cat # C34557 | |
| commercial assay or kit | Celltracker Red CMTPX Dye | Invitrogen | Cat # C34552 | |
| commercial assay or kit | eBioscience Indo-1 AM Calcium Sensor Dye | Invitrogen | Cat # 65-0856-39 | |
| commercial assay or kit | TRIzol Reagent | Invitrogen | Cat # 15596026 | |
| commercial assay or kit | SYBR Green PCR Master Mix | Applied Biosystems | Cat # 4309155 | |
| commercial assay or kit | Mouse TLR1-9 Agonist Kit | Invivogen | Cat # tlrl-kit1mw | |
| commercial assay or kit | Dynabeads sheep anti-rat IgG | Invitrogen | Cat # 11035 | |
| commercial assay or kit | ProLong Gold Antifade Mountant | ThermoFisher | Cat # P36930 | |
| software, algorithm | Flowjo v10.8.1 | BD | https://www.flowjo.com/ | |

*Continued on next page*

*Continued*

| Reagent type (species) or resource | Designation | Source or reference | Identifiers | Additional information |
|---|---|---|---|---|
| software, algorithm | Graphpad PRISM v9.3.1 | Graphpad Software | https://www.graphpad.com/ | |
| software, algorithm | Imaris v9.8.0 | Oxford Instruments | https://imaris.oxinst.com/ | |
| software, algorithm | Biorender | Biorender | https://biorender.com/ | |
| software, algorithm | ImageJ | NIH | https://imagej.nih.gov/ | |

## Mice

C57BL/6J (wild-type, Strain #: 000664), B6.SJL-Ptprc[a] PepC[b]/BoyJ (CD45.1, Strain #: 002014), C57BL/6-Tg(TcraTcrb)1100Mjb/J (OT-I, Strain #: 003831), B6.Cg-Tg(TcraTcrb)425Cbn/J (OT-II, Strain #: 004194), B6(Cg)-Rag2[tm1.1Cgn]/J (*Rag2*[−/−], Strain #: 008449), B6.129P2(C)Ccr7[tm1Rfo]r/J (*Ccr7*[−/−], Strain #: 006621), B6.129S2-H2[dlAb1-Ea]/J (*MHCII*[−/−], Strain #: 003584), B6.129P2-B2m[tm1Unc]/DcrJ (*B2m*[−/−], Strain #: 002087), and C57BL/6-Tg(Nr4a1-EGFP/cre)820Khog/J (*Nr4a1*[GFP], Strain #: 016617) were purchased from the Jackson Laboratory. *Ccr4*[−/−] (*Chvatchko et al., 2000*), pCX-EGFP (*Wright et al., 2001*), and Rag2p-GFP (*Boursalian et al., 2004*) strains were generously provided by A.D. Luster (Massachusetts General Hospital, Boston, MA), Irving L. Weissman (Stanford University, Stanford, CA), and Ellen R. Richie (the University of Texas MD Anderson Center, Houston, TX), respectively. *Ccr4*[−/−]; *Ccr7*[−/−] (DKO), OT-I *Rag2*[−/−], and OT-II *Rag2*[−/−] *MHCII*[−/−] strains were bred in house. Experiments were performed using mice 4–8 weeks old, except for for autoimmune/inflammatory studies which were carried out in older mice, as specified. All strains were bred and maintained under specific pathogen-free conditions at the Animal Resources Center, the University of Texas at Austin, with procedure approval from the Institutional Animal Care and Use Committee, the University of Texas at Austin.

## Flow cytometry

$5 \times 10^6$ thymocytes were stained with fluorescently conjugated antibodies and a viability dye, and incubated on ice for 30 min in the dark. Unless specified, for stains including anti-CCR7, cells were instead incubated in a 37°C water bath for 45 min in the dark. Stained cells were washed and resuspended in FACS Wash Buffer (phosphate-buffered saline [PBS] +2% fetal bovine serum [FBS]; GemCell, Gemini, CA, USA) + 1 µg/ml propidium iodine if a fixable viability dye was not used. For cleaved caspase 3 stains, surface-stained cells were fixed and permeabilized with the BD Cytofix/Cytoperm kit (BD, NJ, USA) according to the manufacturer's instructions prior to staining for anti-cleaved caspase 3 (Cell Signaling Technology, MA, USA). All flow cytometry data were acquired on an LSR Fortessa flow cytometer (BD) or a FACSAria Fusion SORP cell sorter (BD) and analyzed with FlowJo ver.10.8.0 (BD).

## Transwell chemotaxis assays

Thymocyte chemotaxis assays were performed as previously described (*Campbell et al., 1999*). Briefly, $5 \times 10^5$ thymocytes were resuspended in 100 µl RPMI 1640 (Gibco, MA, USA)+10% FBS (Gemini) and added to the top chamber of 5-µm pore transwell tissue culture inserts (Corning, NY, USA) in 24-well plates. The bottom chamber of the transwells contained recombinant mouse chemokines CCL17, CCL22, CCL19, CCL21 (Peprotech, NJ, USA), or CCL25 (R&D Systems, MN, USA) at 100, 10, and 1 nM, diluted in 500 µl RPMI 1640 + 10% FBS. The plate was cultured at 37°C, 5% $CO_2$ for 2 hr. Cells that migrated to the bottom wells, and $5 \times 10^5$ input cells, were analyzed and quantified by flow cytometry. A standard number of 15 µm polyester beads were added to each well to calculate the absolute number of thymocytes that migrated. Migration percentages of each subset were calculated by dividing the number of migrated cells of each subset by the number of cells of the same subset in the input cell sample, and the migration index was calculated as the ratio of the migration percentage in wells containing chemokine to the average migration percentage in wells without chemokine (Blank wells).

## Ex vivo thymic slice preparation

Ex vivo thymic slices were prepared as previous described (*Lancaster and Ehrlich, 2017*). Briefly, surgically removed thymi were cleaned of residual connective tissue, and the two lobes were separated and embedded in 4% low melting point agarose (Lonza, NJ, USA) in PBS. 400 µm thymic slices were generated by slicing the trimmed agarose blocks on a VT 1000S vibratome (Leica, Germany) and kept on ice, submerged in complete RPMI until addition of thymocytes. Slices were placed on 0.4-µm cell culture inserts (Millipore, MA, USA) in 35-mm dishes containing 1 ml complete RPMI (RPMI 1640 supplemented with 1× GlutaMAX, 1 mM sodium pyruvate, 1× penicillin–streptomycin–glutamine, 1× Minimum Essential Medium (MEM) non-essential amino acid, and 50 µM β-mercaptoethanol [All from Gibco], and 10% FBS [Gemini]).

## Synchronized positive selection thymic slice assays

Synchronized positive selection assays were set up as previously described (*Ross et al., 2014*). Briefly, to generate a source of pre-positive selection OT-I thymocytes, bone marrow from OT-I $Rag2^{-/-}$ hosts was magnetically depleted of CD3+, CD8+, CD25+, B220+, Gr-1+, CD11b+, and Ter119+ cells using Rat anti-mouse antibodies (BioXcell, NH, USA) and Dynabeads sheep anti-rat IgG beads (Invitrogen, MA, USA) according to the manufacturer's recommendations, then $10^7$ cells were injected into lethally irradiated $B2m^{-/-}$ hosts. The bone marrow chimera recipients were maintained for 4–6 weeks before collecting OT-I pre-selection thymocytes. For pre-positive selection OT-II thymocytes, OT-II $Rag2^{-/-}$ $MHCII^{-/-}$ mice were bred. Thymic single-cell suspensions from the OT-I $Rag2^{-/-}$ -> $B2m^{-/-}$ bone marrow chimeras or OT-II $Rag2^{-/-}$ $MHCII^{-/-}$ mice were isolated, stained in 5 ml RPMI 1640 + 5 µM CellTrace Violet (Invitrogen; for flow cytometry) or CellTracker RedCMTPX (Thermo Fisher Scientific; for imaging) for 30 min in the dark at 37°C, washed once with warm complete RPMI, resuspended in fresh, warm complete RPMI, and kept in the dark at 37°C prior to slice addition. Prepared cells were overlaid on WT, $B2m^{-/-}$, $MHCII^{-/-}$, or pCX-EGFP slices and incubated at 5% $CO_2$ 37°C. Slices were gently washed at 3 hr with warm complete RPMI in a dish to prevent further entry of new cells. At the indicated time points, thymocytes were harvested by mechanical disruption of the slices for flow cytometry or intact slices were imaged by two-photon microscopy.

## Two-photon imaging and analysis of thymic slices

For imaging of purified thymocyte subsets, sorted cells from each host were centrifuged for collection, then separately stained in RPMI 1640 with 5 µM CellTracker Red CMTPX (Thermo Fisher Scientific, MA, USA) or Indo-1 AM (eBioscience, CA, USA) for 30 min in 37°C water bath, away from light. Dye used for each host was swapped between experiments to account for potential effect of dyes on cell viability and mobility. For imaging of pre-positive selection thymocytes, whole thymus single cells were stained in RPMI 1640 with 5 µM CellTracker Red CMTPX in the same way. Stained cells were collected by centrifugation and resuspended in fresh warm cRPMI, incubated for 30 min in 37°C water bath, away from light. Prepared cells were collected by centrifugation, then washed two times with fresh warm complete RPMI to remove dye residues, then laid onto pCX-EGFP slices, and incubated in 5% $CO_2$ incubator at 37°C for a minimum of 1 hr before two-photon imaging.

For image and video acquisition, thymic slices were secured to an imaging chamber (Harvard Apparatus, MA, USA) perfused with heated DRPMI (Corning) + 2 g/l sodium bicarbonate, 5 mM Hydroxyethylpiperazine Ethane Sulfonic Acid (HEPES) (Sigma-Aldrich, MA, USA) and 1.25 mM $CaCl_2$, pH = 7.4, aerated with 95% $O_2$+ 5% $CO_2$ at flow rate of 100 ml/hr. The imaging chamber is secured on a heated stage, and a temperature probe was inserted to monitor and maintain the chamber temperature at 37°C. Images were acquired using an Ultima IV microscope controlled by PrairieView software (V5.4, Bruker, MA, USA), with a ×20 water immersion objective, NA = 1.0 (Olympus, Japan). Time-lapse videos were acquired by a t-series of 60 rounds of 15 s acquisition, through 40 µm depth at 5 µm intervals. Samples were illuminated with a Mai Tai Ti:sapphire laser (Spectra Physics) tuned to 865 nm for simultaneous excitation of EGFP and CMTPX, and an additional InSight Ti:sapphire laser (Spectra Physics, CA, USA) tuned to 730 nm for excitation of Indo-1 AM, if necessary. Emitted light was passed through 473/24, 525/50, and 605/70 band pass filters (Chroma, VT, USA) to separate PMTs for detecting Indo-1 AM (low calcium), EGFP, and CMTPX signals, respectively.

Captured images were analyzed using Imaris v9.7.2 (Bitplane, Switzerland). For medullary enrichment analysis, medullary and cortical volumes were established by manually generating surface

objects by morphological distinction. Fluorescently labeled cells were identified using Spot tools from random frames of each video, and medullary or cortical cellular localization was determined by calculating the spot's distance to the medullary surface object using the Spot distance to Surface function in ImarisXT. Densities of cells within each region were calculated by dividing the number of cells in the respective region by the volume of the surface object, and medullary-to-cortical ratio was calculated by dividing the medullary density to the cortical density within the same video, using Excel (Microsoft). For thymocyte migration parameters, trajectory of thymocytes were established with the automated Spot tracking feature within Imaris. Only trajectories with time span ≥3 min were included in the analysis. Average track speed and straightness were generated by Imaris for analysis.

## cDNA preparation and qPCR

FACS purified, frozen thymic APC subsets were thawed and lysed in TRIzol (Thermo Fisher), RNA was extracted, and cDNA was synthesized using the qScript cDNA Synthesis Kit (QuantaBio, MA, USA), per manufacturer's recommendations. qRT-PCR was performed using SYBR Green PCR Master Mix (Applied Biosystems, MA, USA), on a ViiA 7 Real-Time PCR System (Applied Biosystems), using the following primers: *Ccl17* Forward: 5'-AGT GGA GTG TTC CAG GGA TG-3', *Ccl17* Reverse: 5'-CCA ATC TGA TGG CCT TCT TC-3', *Ccl19* Forward: 5'-GCT AAT GAT GCG GAA GAC TG-3', *Ccl19* Reverse: 5'-ACT CAC ATC GAC TCT CTA GG-3', *Ccl21* Forward: 5'-GCA GTG ATG AGG GGG TC AG-3', *Ccl21* Reverse: 5'-CGG GGT GAG AAC AGG ATT GC-3', *Ccl22* Forward: 5'-AGG TCC CTA TGG TGC CAA TGT-3', *Ccl22* Reverse: 5'-CGG CAG GAT TTT GAG GTC CA-3', β-actin Forward: 5'-CAC TGT CGA GTC GCG TCC A-3', β-actin Reverse: 5'-CAT CCA TGG CGA ACT GGT GG-3'. ddCT Relative expression levels of target genes in each subset were quantified by first normalizing to actin expression within each subset, then normalizing between subsets for expression of target genes relative to a subset previously shown to express high levels of that gene. For CCL17 and CCL22, expression levels were normalized to cDC2. For CCL19 and CCL21, expression levels were normalized to mTEC$^{lo}$.

## Congenic TLR stimulation and co-culture experiment

To activate APCs, congenic CD45.1 mouse spleen was isolated and red blood cells were lysed with RBC lysis buffer (BioLegend, CA, USA). Leukocytes were then resuspended in complete RPMI with 1 µg/ml high molecular weight poly(I:C), 1 µg/ml low molecular weight poly(I:C), 100 ng/ml LPS, 500 nM ODN1826 (Invivogen, CA, USA), or without any TLR ligands, and incubated in 5% $CO_2$ incubator at 37°C overnight. The next day, CD45.2 WT and *Ccr4*$^{-/-}$ CD4$^+$ T cells were isolated from spleen by magnetically depleting CD8$^+$, CD25$^+$, B220$^+$, CD11b$^+$, Gr-1$^+$, and Ter119$^+$ cells with corresponding rat anti-mouse antibodies (BioXCell) and Dynabeads sheep anti-rat IgG (Invitrogen), and resuspended in complete RPMI. Stimulated CD45.1 splenocytes were washed three times with fresh complete RPMI. $2.5 \times 10^4$ WT or *Ccr4*$^{-/-}$ CD4$^+$ purified T cells were plated with or without $1.25 \times 10^5$ unstimulated or stimulated splenocytes per well in complete RPMI in a 96-well, U-bottom plate. At days 3 and 5, each well was harvested and subjected to flow cytometry analysis. 15-µm polyester beads of known quantities were added to quantify cells in each sample.

## Immunofluorescent analyses of thymic and LNs cryosections and detection of autoantibodies

To detect anti-nuclear autoantibodies in mouse serum, 7 µm cryosections were prepared from *Rag2*$^{-/-}$ kidneys (C57BL/6J background). Cryosections were fixed in acetone (–20°C for 20 min), rinsed 3× (2× PBS rinses and 1× PBS with 0.1% Tween 20), blocked with 9% donkey serum (Jackson ImmunoResearch) in PBS for 20 min and incubated with undiluted mouse serum from mice of the indicated genotypes for 2 hr at room temperature. After washing, auto-antibodies were detected by staining with anti-mouse IgG AF594 in PBS for 1 hr at room temperature. After washing, slides were incubated with 4',6-diamidino-2-phenylindole (DAPI), washed, and coverslips were mounted in ProLong Gold Antifade reagent (Thermo Fisher). For CCL22 immunofluorescent staining in thymus and inguinal lymph node immunostaining, 7 µm cryosections were prepared from the thymus of 1-month-old C57BL/6J mice and from lymph nodes of 5- to 6-month-old mice, respectively. All slides were fixed and stained as above. Thymus sections were stained with anti-CCL22, anti-CD11c-biotin, and anti-CD31-APC overnight at room temperature. After washing, sections were incubated with anti-goat-Dylight 488 and streptavidin Alexa Fluor 594 secondary antibodies for 1hr at room temperature. Inguinal lymph

nodes were stained with anti-CD8a-AF594, anti-B220 -biotin, and anti-CD4-APC for 4 hours at room temperature. After washing, sections were incubated with streptavidin Alexa Fluor 488 for 1hr at room temperature. After washing, the DAPI nuclear stain was added and slides were mounted as above. Immunofluorescent images were acquired on a DMi8 microscope (Leica), using a ×10/0.4 NA objective or a ×20/0.7 NA objective. Stitched images were generated with LasX software (Leica). All images were uniformly processed and converted into Tiffs using Fiji software (ImageJ).

## H&E staining and quantification

Lacrimal glands, submandibular glands, colon, and liver were extracted from 6- to 8-month-old mice and fixed in formalin (Fisherbrand) for 48 hr before storing in 70% ethanol. Organs were sent to the Histology and Immunohistochemistry Laboratory at the University of Texas San Antonio for embedding, sectioning, and staining. Spleen and mLNs were extracted and processed by the Histology and Tissue Processing core at the University of Texas MD Anderson Cancer Center Science Park (Smithville, TX). Histological analysis of spleen, inguinal lymph nodes, liver, and colon was done by a veterinary pathologist, while infiltrates in lacrimal and submandibular glands stains were analyzed in-house, as previously described (*Lieberman et al., 2015*). Briefly, we scored foci composed of at least 50 mononuclear cells. In the cases where multiple foci coalesced, we assigned foci a score value of 3 for statistical analysis. The number of inflammatory foci per 10 mm$^2$ was calculated by counting the total number of foci by standard light microscopy using a ×10 objective and dividing that by the surface area of sections measured by Fiji software. Colon sections were scored by Dr. Hale, who was blinded to genotype, for mucosal changes such as hyperplasia and architectural distortion, inflammation severity, and the percentage of the tissue affected as described previously (*Hale et al., 2005*), but were given a single score that reflected the entire segment examined (total possible scores ranged from 0 to 15).

## Statistical analysis

All statistical analysis was conducted using GraphPad PRISM v9.8 (GraphPad Software, CA, USA), with the corresponding statistical tests and multiple comparison corrections listed in the figure legends.

## Acknowledgements

The authors thank the Animal Resource Center staff at the University of Texas at Austin for assistance with mouse maintenance, Dr. Jessica Lancaster for assistance with two-photon microscopy training, Richard Salinas at the Center for Biomedical Research Support, and Dr. Ellen Richie for providing advice. Graphical illustrations were created with Biorender.com. This research was supported by a grant from the National Institutes of Health R01AI104870 to LIRE.

## Additional information

### Funding

| Funder | Grant reference number | Author |
| --- | --- | --- |
| National Institutes of Health | R01AI104870 | Lauren IR Ehrlich |

The funders had no role in study design, data collection, and interpretation, or the decision to submit the work for publication.

### Author contributions

Yu Li, Conceptualization, Formal analysis, Investigation, Writing – original draft, Writing – review and editing; Pablo Guaman Tipan, Formal analysis, Investigation, Writing – original draft, Writing – review and editing; Hilary J Selden, Investigation, Project administration; Jayashree Srinivasan, Formal analysis, Investigation; Laura P Hale, Formal analysis, Writing – original draft; Lauren IR Ehrlich, Conceptualization, Formal analysis, Supervision, Funding acquisition, Writing – original draft, Writing – review and editing

## Author ORCIDs

Yu Li ![ORCID] http://orcid.org/0000-0002-1893-9046
Pablo Guaman Tipan ![ORCID] http://orcid.org/0009-0007-3127-7095
Jayashree Srinivasan ![ORCID] http://orcid.org/0000-0001-9478-7518
Lauren IR Ehrlich ![ORCID] http://orcid.org/0000-0002-1697-1755

## Ethics

This study was performed in strict accordance with the recommendations in the Guide for the Care and Use of Laboratory Animals of the National Institutes of Health. All of the animals were handled according to approved Institutional Animal Care and Use Committee (IACUC) protocol (# AUP-2019-00034) at the University of Texas at Austin.

## Decision letter and Author response

Decision letter https://doi.org/10.7554/eLife.80443.sa1
Author response https://doi.org/10.7554/eLife.80443.sa2

## Additional files

### Supplementary files
- MDAR checklist
- Source data 1. Compiled data used to generate graphs in all figures and figure supplements.

### Data availability

All data in this study are included in *Source data 1*. Data for individual components of figures are found on the corresponding tabs in the spreadsheet.

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
