## [Editor Report]

This important paper reveals the key steps associated with intrathymic central tolerance. Using elegant live imaging approaches, the authors provide convincing evidence in support of an updated model for how positive-selected thymocytes are called into the thymus medulla to interact with distinct antigen-presenting cells. The work makes an important contribution to the field by identifying previously unappreciated complexities related to cellular movement during T cell generation.

---

## [Decision Letter]

**Decision letter after peer review:**

Thank you for submitting your article "CCR4 and CCR7 differentially regulate thymocyte localization with distinct outcomes for central tolerance" for consideration by *eLife*. Your article has been reviewed by 3 peer reviewers, including Juan Carlos Zúñiga-Pflücker as the Reviewing Editor and Reviewer #1, and the evaluation has been overseen by Tadatsugu Taniguchi as the Senior Editor. The following individuals involved in the review of your submission have agreed to reveal their identity: Yousuke Takahama (Reviewer #2).

Essential revisions:

1) It was felt that a more focused manuscript examining CCR4 expression and function would increase the impact of the work. In particular, lessening the reporting of previously established expression and function of CCR7.

2) More careful description and/or interpretation of the results from the caspase-3 analysis are encouraged, as well as, whether thymocytes make direct contact with CCR4-ligand expressing cells.

3) Please carefully consider and address the concerns raised by the reviewers.

*Reviewer #1 (Recommendations for the authors):*

One outstanding question that remains is whether mice that are deficient for both CCR4 and CCR7, or even CCR4 alone, would show an increased incidence of spontaneous autoimmunity or auto-inflammatory outcomes. Have the author's noted or looked for any evidence of whether a break in tolerance leads to pathological consequences. This would nicely support their findings in Figure 7.

*Reviewer #2 (Recommendations for the authors):*

Figure 5C: All the density data look smaller in CCR7-/- than WT. Paired comparison between WT and CCR7-/- data should be additionally shown. The role of CCR4 is better evaluated by comparing the impacts of CCR4 and CCR7.

Figure 6C, D: These experiments measured caspase 3 cleavage in thymocyte subpopulations. Caspase 3 cleavage reflects apoptosis of thymocytes induced by various signals including glucocorticoid hormones (Alam, et al. J Exp Med. 1997 and Marchetti, et al. Blood 2003). It was also reported that caspase 3 inhibition does not always inhibit the negative selection of thymocytes (Doerfler, et al. J Immunol. 2000). Thus, the figure panels should specify the measurement of caspase 3 cleavage, and the text should convey a careful interpretation of the data with the biology of negative selection.

Figure 7A, B: Some DCs, including cDC2, are reported to be localized in the cortex within the thymus (Baba, et al., J Immunol. 2009). Also, thymocyte-DC interactions in the cortex are visualized to be CCR7-dependent (Ladi, et al. J Immunol. 2008). Whether cDC2 and CCR7+ cDCs are localized in the thymic cortex or the medulla is important to verify the possibility that CCR4 promotes the medullary migration of thymocytes. CCR4 may be more important for thymocyte-DC interaction in the cortex for the early phase of negative selection than promoting thymocyte migration to the medulla.

*Reviewer #3 (Recommendations for the authors):*

1. In Figure 6, the authors examine 'early-phase' and 'late-phase' clonal deletion in various mice. Measurement of 'early-phase' clonal deletion does not appear to be impacted by combined CCR4/CCR7 deficiency. Can the authors comment on this finding?

2. Generally, the impact of CCR4 signaling on thymocyte 'outputs' is modest in comparison to CCR7 signaling (migration – Figure 2, medullary accumulation – Figure 5). This doesn't appear to be discussed. Is this due to the stage of the thymocyte stage and that TCR signaled DP and SM thymocytes are less responsive generally or do the authors think there are other factors? This may be pertinent to the authors' findings of reduced responsiveness of more immature, TCR signaled cells to CXCR4 ligands.

3. Throughout the manuscript the authors refer to DP CD3lo CD69+ cells and DP CD3+ CD69+ cells as post-positive selection. In a polyclonal repertoire, separating "TCR signaled" (CD69+) thymocytes into those that received positive selection signals versus negative selection signals is difficult. This description of 'post-positive selection' seems inaccurate and unsubstantiated. Have the authors examined Helios expression or expression of other markers (Nur77) to bolster their descriptions?

4. This is a minor point, but when the authors are discussing the cell populations present/absent in b2m-/- mice, there are obviously cells restricted by non-classical MHC-I as well as classical MHC-I that would be impacted. This should be appropriately considered (lines 102-118 for instance).

5. In the legend of figure 5 (or figure 5 itself), the time point of analysis does not appear to be stated.

---

## [Author Response]

Essential revisions:Reviewer #1 (Recommendations for the authors):One outstanding question that remains is whether mice that are deficient for both CCR4 and CCR7, or even CCR4 alone, would show an increased incidence of spontaneous autoimmunity or auto-inflammatory outcomes. Have the author's noted or looked for any evidence of whether a break in tolerance leads to pathological consequences. This would nicely support their findings in Figure 7.

We thank the reviewer for this suggestion. We agree that an important implication of our model is that deficiency in CCR4 and/or CCR7 could result in an increased risk of spontaneous autoimmunity, which we have assessed in additional experiments. The link between CCR7 deficiency and autoimmunity has been well documented (Davalos-Misslitz, Rieckenberg, et al., 2007; Davalos-Misslitz, Worbs, et al., 2007; Kozai et al., 2017; Kurobe et al., 2006), and our previous study showed that CCR4 deficiency leads to increased lymphocytic infiltrates in lacrimal glands and circulating anti-nuclear autoantibodies in mice approaching 12-months of age (Hu et al., 2015). We’ve now highlighted these results in our revised text. In addition, because we find that CCR4 and CCR7 synergize to promote polyclonal negative selection and individually impact tolerance of different stages of developing thymocytes (Figure 6C), we evaluated whether distinct autoimmune manifestations would emerge in *Ccr4^-/-^, Ccr7^-/-^*, and *Ccr4^-/^ Ccr7^-/-^* mice at 4.5-6.5 months of age. Notably, severe inflammation in the colon was evident only in *Ccr4^-/^ Ccr7^-/-^* mice, with a trend towards epithelial hyperplasia in *Ccr4*^-/-^ mice (Figure 7A). *Ccr7* deficiency and double deficiency resulted in lymphocytic infiltrates in endocrine organs and liver inflammation, consistent with a more prominent role for CCR7 in sustaining tolerance to tissue-specific antigens (Figure 7B and Figure 7—figure supplement 1B-D). In addition, lymphoid hyperplasia was observed in secondary lymphoid organs of *Ccr4^-/-^* but not *Ccr7^-/-^* mice (Figure 7C), and disorganized T cell zones were evident in *Ccr4*^-/-^ lymph nodes (Figure 7C). These results are consistent with our finding that CCR4 induces T-cell tolerance to self-antigens displayed by professional antigen presenting cells (Figure 7F). We discuss these findings and their implications in our revised Results and Discussion.

Reviewer #2 (Recommendations for the authors):Figure 5C: All the density data look smaller in CCR7-/- than WT. Paired comparison between WT and CCR7-/- data should be additionally shown. The role of CCR4 is better evaluated by comparing the impacts of CCR4 and CCR7.

We thank the reviewer for this sharp observation. As requested, we have carried out additional 2-photon imaging experiments, pairing sorted WT and *Ccr7^-/-^* thymocyte subsets. These paired analyses reveal that CCR7 deficiency results in a decrease in medullary enrichment of CD4SP SM and M1+M2 subsets, while no significant difference was observed for DP CD3^lo^CD69^+^ subset (Figure 5C). We also added data from these new imaging experiments to the dataset graphed in Figure 5B. Despite the difference revealed via paired analyses, the compiled data continue to indicate that CD4SP SM cells rely on a combination of CCR4 and CCR7 signaling to accumulate in the medulla, and only single CCR4 deficiency significantly impairs medullary entry of the CD4SP SM subset. Additional new imaging data in Figure 5B also confirm that CCR4 is uniquely required for medullary accumulation of post-positive selection DP CD3^lo^CD69^+^ cells, as both *Ccr4*^-/-^ and *Ccr4^-/-^Ccr7^-/-^* cells distribute evenly with a medullary:cortical density ration of ~1. We now incorporate these additional data in our manuscript and think these enhanced findings strengthen our conclusion about the critical role of CCR4 in inducing early medullary entry and CCR7 in inducing late medullary entry.

Figure 6C, D: These experiments measured caspase 3 cleavage in thymocyte subpopulations. Caspase 3 cleavage reflects apoptosis of thymocytes induced by various signals including glucocorticoid hormones (Alam, et al. J Exp Med. 1997 and Marchetti, et al. Blood 2003). It was also reported that caspase 3 inhibition does not always inhibit the negative selection of thymocytes (Doerfler, et al. J Immunol. 2000). Thus, the figure panels should specify the measurement of caspase 3 cleavage, and the text should convey a careful interpretation of the data with the biology of negative selection.

We thank the reviewer for the suggestion. It has been established in multiple previous studies that Cleaved Caspase 3 in TCR-signaled thymocytes is an indicator mainly of negative selection (Breed et al., 2019; Hu et al., 2016). However, as the reviewer points out, the intracellular cleaved caspase 3 could reflect apoptosis triggered by other mechanisms as well. Figure 6C has been modified to reflect the measurement of % Cleaved Caspase 3^+^ on the graph axes and title, and we have modified our text to acknowledge that while apoptosis of TCR-signaled post-positive selection Ccasp3^+^ thymocytes is consistent with detection of negative selection, it could also reflect some apoptosis induced by additional signals like glucocorticoids.

Figure 7A, B: Some DCs, including cDC2, are reported to be localized in the cortex within the thymus (Baba, et al., J Immunol. 2009). Also, thymocyte-DC interactions in the cortex are visualized to be CCR7-dependent (Ladi, et al. J Immunol. 2008). Whether cDC2 and CCR7+ cDCs are localized in the thymic cortex or the medulla is important to verify the possibility that CCR4 promotes the medullary migration of thymocytes. CCR4 may be more important for thymocyte-DC interaction in the cortex for the early phase of negative selection than promoting thymocyte migration to the medulla.

We thank the reviewer for this comment. While the majority of CD11c^+^ DCs are enriched in the medulla, some are present in the cortex. Our data show that thymic cDCs do not express CCR7 ligands, and instead CCR7 ligands are expressed by mTECs (Figure 7E), consistent with previous reports (Ki et al., 2014; Kozai et al., 2017). These data are not consistent with CCR7-dependent thymocyte-DC interactions in the cortex. To investigate whether CCR4 ligands constitute a chemokine gradient that would promote chemotaxis towards the medulla, we performed a CCL22 immunofluorescent stain on thymic sections (Figure 2—figure supplement 2) and found that CCL22 protein is concentrated in the medulla, similar to the distribution of CCL19 and CCL21 in the thymic medulla (Kozai et al., 2017; Ueno et al., 2002; Ueno et al., 2004). Therefore, the gradient of CCL22 in the medulla is consistent with CCR4-mediated chemotaxis of post-positive selection thymocytes subsets into the medulla, which we now discuss. However, we also modified the text to acknowledge that CCR4 could induce interactions between post-positive selection thymocytes and sparse DCs in the cortex, which is particularly interesting as both DP CD3^lo^CD69^+^ and CD4SP SM cells respond to CCR4 ligands and migrate both in the cortex and medulla.

Reviewer #3 (Recommendations for the authors):1. In Figure 6, the authors examine 'early-phase' and 'late-phase' clonal deletion in various mice. Measurement of 'early-phase' clonal deletion does not appear to be impacted by combined CCR4/CCR7 deficiency. Can the authors comment on this finding?

We thank the reviewer for raising this point; a subtle, but not significant decrease in early-phase negative selection was observed between WT and *Ccr4^-/-^Ccr7^-/-^* DKO mice in the original figure. To further test whether double-deficiency does impact early-phase negative selection, as expected by our model, we analyzed additional *Ccr4^-/-^Ccr7^-/-^* mice. These additional data demonstrate that combined CCR4 and CCR7 deficiency induces a statistically significant decrease in early-phase negative selection (Figure 6C). This result solidified our findings that CCR4 is important for early-phase negative selection, and we now discuss these updated data in our revised manuscript.

2. Generally, the impact of CCR4 signaling on thymocyte 'outputs' is modest in comparison to CCR7 signaling (migration – Figure 2, medullary accumulation – Figure 5). This doesn't appear to be discussed. Is this due to the stage of the thymocyte stage and that TCR signaled DP and SM thymocytes are less responsive generally or do the authors think there are other factors? This may be pertinent to the authors' findings of reduced responsiveness of more immature, TCR signaled cells to CXCR4 ligands.

We thank the reviewer for this insight. It is possible that the different magnitude of chemotactic responses induced by CCR4, CCR7 and CXCR4 could reflect intrinsic differences in responsiveness of immature and mature thymocytes to chemokine signals, as suggested by the much lower migration of CD4SP SM relative to CD4SP M1 and M2 to CCR7 ligands. To address this possibility, we investigated the ability of thymocyte subsets to undergo chemotaxis to CCL25, the ligand for CCR9, which has been shown to induce chemotaxis in all but the most mature thymocyte subsets (Campbell, Pan, et al., 1999). We found that CD4SP SM cells migrated as robustly to CCL25 as both earlier DP CD3^lo^CD69^+^ cells and later CD4SP M1 cells, indicating that CD4SPSM cells do not have an inherently inefficient ability to respond to chemotactic cues (Figure 2—figure supplement 2). These data are also consistent with migration of CD4SP SM cells to CCR4 ligands at near maximal levels (Figure 2B). Therefore, it is unlikely that CD4SP SM cells are unresponsive to all chemokine signals, but are rather insensitive to specific chemokine stimulation, as seen by their lack of robust responses to CXCR4 or CCR7 ligands (Figure 2D and Figure 3), despite strong expression of these chemokine receptors by this subset. We now discuss these additional findings. While resolving why this CD4SP SM subset responds to some chemokines, but not others for which receptors are expressed is of great interest, and will further our understanding of how chemokine receptors orchestrate thymocyte movement and selection in the thymus, it is beyond the scope of this study, but worthy of further investigation.

3. Throughout the manuscript the authors refer to DP CD3lo CD69+ cells and DP CD3+ CD69+ cells as post-positive selection. In a polyclonal repertoire, separating "TCR signaled" (CD69+) thymocytes into those that received positive selection signals versus negative selection signals is difficult. This description of 'post-positive selection' seems inaccurate and unsubstantiated. Have the authors examined Helios expression or expression of other markers (Nur77) to bolster their descriptions?

The reviewer raises an interesting point that the observed DP CD3^lo^CD69^+^ and DP CD3^+^CD69^+^ thymocytes may not be representative of post-positive selection cells, but rather cells that could be signaled through the TCR to undergo negative selection at the DP stage. Previous studies showed that thymocytes undergoing signaling that drives negative selection exhibit TCR signaling strength similar to endogenous self-reactive Tregs, measured by Nur77-GFP reporter levels (Stritesky et al., 2013). Thus, to test whether DP CD3^lo^CD69^+^ cells reflect mainly cells that have received positive selection TCR signals, versus cells undergoing TCR signaling reflecting negative selection, we compared Nur77-GFP reporter levels of DP CD3^lo^CD69^+^ and DP CD3^+^CD69^+^ thymocytes with that of thymic Treg precursors (Figure 1—figure supplement 1). Levels of Nur77-GFP in both DP subsets were substantially lower than in Tregs, indicating that these cells largely do not reflect cells undergoing high-avidity TCR signaling associated with negative selection. This finding is consistent with the low frequency, ~ 0.3%, of DP CD69^+^ cells that are undergoing apoptosis, reflected by cleaved caspase 3 expression (Figure 6C). We have now modified the text to discuss this point.

4. This is a minor point, but when the authors are discussing the cell populations present/absent in b2m-/- mice, there are obviously cells restricted by non-classical MHC-I as well as classical MHC-I that would be impacted. This should be appropriately considered (lines 102-118 for instance).

We thank the reviewer for the insight. Development of non-classical MHC-I restricted thymocytes, including iNKT cells and CD8αα+ IELps, are disrupted in *β2m^-/-^* mice (Bendelac et al., 1994; Ruscher et al., 2017). We have modified the manuscript as suggested to acknowledge this point.

5. In the legend of figure 5 (or figure 5 itself), the time point of analysis does not appear to be stated.

We apologize for this omission and have clarified in the figure legend that the slices were imaged between 1-4 hours after introduction of thymocytes. We did not observe any significant changes in the efficiency of thymocyte accumulation in the medulla within this time frame when analyzing localization of sorted subsets.

References:

Au-Yeung, B. B., Melichar, H. J., Ross, J. O., Cheng, D. A., Zikherman, J., Shokat, K. M., Robey, E. A., and Weiss, A. (2014). Quantitative and temporal requirements revealed for Zap70 catalytic activity during T cell development. *Nat Immunol*, *15*(7), 687-694. https://doi.org/10.1038/ni.2918

Bendelac, A., Killeen, N., Littman, D. R., and Schwartz, R. H. (1994). A subset of CD4^+^ thymocytes selected by MHC class I molecules. *Science*, *263*(5154), 1774-1778. https://doi.org/10.1126/science.7907820

Breed, E. R., Watanabe, M., and Hogquist, K. A. (2019). Measuring Thymic Clonal Deletion at the Population Level. *J Immunol*, *202*(11), 3226-3233. https://doi.org/10.4049/jimmunol.1900191

Britschgi, M. R., Link, A., Lissandrin, T. K. A., and Luther, S. A. (2008). Dynamic Modulation of CCR7 Expression and Function on Naive T Lymphocytes in vivo1. *The Journal of Immunology*, *181*(11), 7681-7688. https://doi.org/10.4049/jimmunol.181.11.7681

Campbell, J. J., Haraldsen, G., Pan, J., Rottman, J., Qin, S., Ponath, P., Andrew, D. P., Warnke, R., Ruffing, N., Kassam, N., Wu, L., and Butcher, E. C. (1999). The chemokine receptor CCR4 in vascular recognition by cutaneous but not intestinal memory T cells. *Nature*, *400*(6746), 776-780. https://doi.org/10.1038/23495

Campbell, J. J., Pan, J., and Butcher, E. C. (1999). Cutting edge: developmental switches in chemokine responses during T cell maturation. *J Immunol*, *163*(5), 2353-2357. https://doi.org/ji_v163n5p2353 [pii]

Cowan, J. E., McCarthy, N. I., Parnell, S. M., White, A. J., Bacon, A., Serge, A., Irla, M., Lane, P. J., Jenkinson, E. J., Jenkinson, W. E., and Anderson, G. (2014). Differential requirement for CCR4 and CCR7 during the development of innate and adaptive alphabetaT cells in the adult thymus. *J Immunol*, *193*(3), 1204-1212. https://doi.org/10.4049/jimmunol.1400993

Davalos-Misslitz, A. C., Rieckenberg, J., Willenzon, S., Worbs, T., Kremmer, E., Bernhardt, G., and Forster, R. (2007). Generalized multi-organ autoimmunity in CCR7-deficient mice. *Eur J Immunol*, *37*(3), 613-622. https://doi.org/10.1002/eji.200636656

Davalos-Misslitz, A. C., Worbs, T., Willenzon, S., Bernhardt, G., and Forster, R. (2007). Impaired responsiveness to T-cell receptor stimulation and defective negative selection of thymocytes in CCR7-deficient mice. *Blood*, *110*(13), 4351-4359. https://doi.org/10.1182/blood-2007-01-070284

Doerfler, P., Forbush, K. A., and Perlmutter, R. M. (2000). Caspase enzyme activity is not essential for apoptosis during thymocyte development. *J Immunol*, *164*(8), 4071-4079. https://doi.org/10.4049/jimmunol.164.8.4071

Dzhagalov, I. L., Chen, K. G., Herzmark, P., and Robey, E. A. (2013). Elimination of self-reactive T cells in the thymus: a timeline for negative selection. *PLoS Biol*, *11*(5), e1001566. https://doi.org/10.1371/journal.pbio.1001566

Ehrlich, L. I., Oh, D. Y., Weissman, I. L., and Lewis, R. S. (2009). Differential contribution of chemotaxis and substrate restriction to segregation of immature and mature thymocytes. *Immunity*, *31*(6), 986-998. https://doi.org/10.1016/j.immuni.2009.09.020

Hu, D. Y., Yap, J. Y., Wirasinha, R. C., Howard, D. R., Goodnow, C. C., and Daley, S. R. (2016). A timeline demarcating two waves of clonal deletion and Foxp3 upregulation during thymocyte development. *Immunol Cell Biol*, *94*(4), 357-366. https://doi.org/10.1038/icb.2015.95

Hu, Z., Lancaster, J. N., Sasiponganan, C., and Ehrlich, L. I. (2015). CCR4 promotes medullary entry and thymocyte-dendritic cell interactions required for central tolerance. *J Exp Med*, *212*(11), 1947-1965. https://doi.org/10.1084/jem.20150178

Kadakia, T., Tai, X., Kruhlak, M., Wisniewski, J., Hwang, I. Y., Roy, S., Guinter, T. I., Alag, A., Kehrl, J. H., Zhuang, Y., and Singer, A. (2019). E-protein-regulated expression of CXCR4 adheres preselection thymocytes to the thymic cortex. *J Exp Med*, *216*(8), 1749-1761. https://doi.org/10.1084/jem.20182285

Ki, S., Park, D., Selden, H. J., Seita, J., Chung, H., Kim, J., Iyer, V. R., and Ehrlich, L. I. R. (2014). Global transcriptional profiling reveals distinct functions of thymic stromal subsets and age-related changes during thymic involution. *Cell Rep*, *9*(1), 402-415. https://doi.org/10.1016/j.celrep.2014.08.070

Kimura, M. Y., Thomas, J., Tai, X., Guinter, T. I., Shinzawa, M., Etzensperger, R., Li, Z., Love, P., Nakayama, T., and Singer, A. (2016). Timing and duration of MHC I positive selection signals are adjusted in the thymus to prevent lineage errors. *Nat Immunol*, *17*(12), 1415-1423. https://doi.org/10.1038/ni.3560

Kozai, M., Kubo, Y., Katakai, T., Kondo, H., Kiyonari, H., Schaeuble, K., Luther, S. A., Ishimaru, N., Ohigashi, I., and Takahama, Y. (2017). Essential role of CCL21 in establishment of central self-tolerance in T cells. *J Exp Med*, *214*(7), 1925-1935. https://doi.org/10.1084/jem.20161864

Kurobe, H., Liu, C., Ueno, T., Saito, F., Ohigashi, I., Seach, N., Arakaki, R., Hayashi, Y., Kitagawa, T., Lipp, M., Boyd, R. L., and Takahama, Y. (2006). CCR7-dependent cortex-to-medulla migration of positively selected thymocytes is essential for establishing central tolerance. *Immunity*, *24*(2), 165-177. https://doi.org/10.1016/j.immuni.2005.12.011

Kwan, J., and Killeen, N. (2004). CCR7 directs the migration of thymocytes into the thymic medulla. *J Immunol*, *172*(7), 3999-4007. https://doi.org/10.4049/jimmunol.172.7.3999

Lancaster, J. N., Thyagarajan, H. M., Srinivasan, J., Li, Y., Hu, Z., and Ehrlich, L. I. R. (2019). Live-cell imaging reveals the relative contributions of antigen-presenting cell subsets to thymic central tolerance. *Nat Commun*, *10*(1), 2220. https://doi.org/10.1038/s41467-019-09727-4

Lutes, L. K., Steier, Z., McIntyre, L. L., Pandey, S., Kaminski, J., Hoover, A. R., Ariotti, S., Streets, A., Yosef, N., and Robey, E. A. (2021). T cell self-reactivity during thymic development dictates the timing of positive selection. *ELife*, *10*, 1-28. https://doi.org/10.7554/*eLife*.65435

Melichar, H. J., Ross, J. O., Herzmark, P., Hogquist, K. A., and Robey, E. A. (2013). Distinct temporal patterns of T cell receptor signaling during positive versus negative selection in situ. *Sci Signal*, *6*(297), ra92. https://doi.org/10.1126/scisignal.2004400

Nitta, T., Nitta, S., Lei, Y., Lipp, M., and Takahama, Y. (2009). CCR7-mediated migration of developing thymocytes to the medulla is essential for negative selection to tissue-restricted antigens. *Proc Natl Acad Sci U S A*, *106*(40), 17129-17133. https://doi.org/10.1073/pnas.0906956106

Ross, J. O., Melichar, H. J., Au-Yeung, B. B., Herzmark, P., Weiss, A., and Robey, E. A. (2014). Distinct phases in the positive selection of CD8^+^ T cells distinguished by intrathymic migration and T-cell receptor signaling patterns. *Proc Natl Acad Sci U S A*, *111*(25), E2550-2558. https://doi.org/10.1073/pnas.1408482111

Ruscher, R., Kummer, R. L., Lee, Y. J., Jameson, S. C., and Hogquist, K. A. (2017). CD8alphaalpha intraepithelial lymphocytes arise from two main thymic precursors. *Nat Immunol*, *18*(7), 771-779. https://doi.org/10.1038/ni.3751

Saini, M., Sinclair, C., Marshall, D., Tolaini, M., Sakaguchi, S., and Seddon, B. (2010). Regulation of Zap70 expression during thymocyte development enables temporal separation of CD4 and CD8 repertoire selection at different signaling thresholds. *Sci Signal*, *3*(114), ra23. https://doi.org/10.1126/scisignal.2000702

Stritesky, G. L., Xing, Y., Erickson, J. R., Kalekar, L. A., Wang, X., Mueller, D. L., Jameson, S. C., and Hogquist, K. A. (2013). Murine thymic selection quantified using a unique method to capture deleted T cells. *Proc Natl Acad Sci U S A*, *110*(12), 4679-4684. https://doi.org/10.1073/pnas.1217532110

Ueno, T., Hara, K., Willis, M. S., Malin, M. A., Hopken, U. E., Gray, D. H., Matsushima, K., Lipp, M., Springer, T. A., Boyd, R. L., Yoshie, O., and Takahama, Y. (2002). Role for CCR7 ligands in the emigration of newly generated T lymphocytes from the neonatal thymus. *Immunity*, *16*(2), 205-218. https://doi.org/10.1016/s1074-7613(02)00267-4

Ueno, T., Saito, F., Gray, D. H., Kuse, S., Hieshima, K., Nakano, H., Kakiuchi, T., Lipp, M., Boyd, R. L., and Takahama, Y. (2004). CCR7 signals are essential for cortex-medulla migration of developing thymocytes. *J Exp Med*, *200*(4), 493-505. https://doi.org/10.1084/jem.20040643